# G-protein-coupled receptor P2Y10 facilitates chemokine-induced CD4 T cell migration through autocrine/paracrine mediators

Malarvizhi Gurusamy[1], Denise Tischner[1], Jingchen Shao[1], Stephan Klatt [2], Sven Zukunft [2], Remy Bonnavion [1], Stefan Günther[3], Kai Siebenbrodt[4], Roxane-Isabelle Kestner[4], Tanja Kuhlmann [5], Ingrid Fleming[2], Stefan Offermanns [1,6] & Nina Wettschureck [1,6✉]

G-protein-coupled receptors (GPCRs), especially chemokine receptors, play a central role in the regulation of T cell migration. Various GPCRs are upregulated in activated CD4 T cells, including P2Y10, a putative lysophospholipid receptor that is officially still considered an orphan GPCR, i.e., a receptor with unknown endogenous ligand. Here we show that in mice lacking P2Y10 in the CD4 T cell compartment, the severity of experimental autoimmune encephalomyelitis and cutaneous contact hypersensitivity is reduced. P2Y10-deficient CD4 T cells show normal activation, proliferation and differentiation, but reduced chemokine-induced migration, polarization, and RhoA activation upon in vitro stimulation. Mechanistically, CD4 T cells release the putative P2Y10 ligands lysophosphatidylserine and ATP upon chemokine exposure, and these mediators induce P2Y10-dependent RhoA activation in an autocrine/paracrine fashion. ATP degradation impairs RhoA activation and migration in control CD4 T cells, but not in P2Y10-deficient CD4 T cells. Importantly, the P2Y10 pathway appears to be conserved in human T cells. Taken together, P2Y10 mediates RhoA activation in CD4 T cells in response to auto-/paracrine-acting mediators such as LysoPS and ATP, thereby facilitating chemokine-induced migration and, consequently, T cell-mediated diseases.

[1] Department of Pharmacology, Max Planck Institute for Heart and Lung Research, Bad Nauheim, Germany. [2] Centre of Molecular Medicine, Goethe University, Frankfurt am Main, Germany. [3] Deep sequencing platform, Max Planck Institute for Heart and Lung Research, Bad Nauheim, Germany. [4] Department of Neurology, University of Frankfurt, Frankfurt, Germany. [5] Institute of Neuropathology, University of Münster, Münster, Germany. [6] Medical Faculty, Goethe University Frankfurt, Frankfurt, Germany. ✉email: Nina.Wettschureck@mpi-bn.mpg.de

Coordinated leukocyte migration between lymphatic organs or towards inflammatory stimuli is crucial for adequate immune responses. G-protein-coupled receptors (GPCRs) are well known regulators of leukocyte migration, they allow cells to respond to a wide range of chemoattractants such as chemokines[1], lysophospholipids[2,3], or prostanoids[4,5]. However, these three groups of receptors represent only a small fraction of known GPCRs, and a systematic analysis of GPCR expression in murine lymphoid organs revealed that approximately 173 GPCRs are expressed in spleen, thymus, and bone marrow[6], among them many orphan GPCRs, i.e., GPCRs for which neither ligand nor function is known[7–9]. We recently analyzed GPCR expression at the single-cell level during neuroinflammation and identified a number of GPCRs that are strongly upregulated in spinal cord infiltrating CD4 cells, among them the class A orphan GPCR P2Y10[10], which so far has not been studied in T cells.

P2Y10 was first described as a lymphoid-restricted receptor regulated by the Ets factors PU.1 and Spi-B in B cells[11], but it is also expressed in subsets of T cells, monocytes, dendritic cells, and granulocytes[12]. Based on sequence homology, P2Y10 was originally thought to be a nucleotide receptor of the purinergic P2Y family (see annotation for GenBank file AF000545.1), but later phylogenetic analyses found it rather related to the receptor for platelet-activating factor[13]. Subsequently, studies in transfected CHO cells indicated that P2Y10 mediates calcium mobilization in response to lysophosphatidic acid (LPA) and sphingosine-1-phosphate (S1P)[14], but this finding was not confirmed by others[15]. Using a TGFα shedding assay, P2Y10 was then suggested as a receptor for lyso-phosphatidylserine (LysoPS), a bioactive lipid that activates also several other putative lysophospholipid receptors such as GPR34, GPR174, and P2Y10b (also known as A630033H20Rik)[16,17]. However, since none of the putative P2Y10 ligands were unanimously confirmed, the International Union of Basic & Clinical Pharmacology still lists P2Y10 as an orphan GPCR (https://www.guidetopharmacology.org). Also, the intracellular signalling cascades employed by P2Y10 are poorly understood, though a coupling to the $G_{12/13}$ family G-proteins has been suggested[18]. Regarding its biological function, indirect data point to a role of P2Y10 in eosinophil degranulation[12], and in microglia and dendritic cells, P2Y10 was implicated in oleamide-mediated suppression of inflammatory responses[19]. In murine neuroinflammation, P2Y10 is strongly upregulated, especially in spinal cord-infiltrating CD4 T cells, but its function in neuroinflammation is not known.

Multiple sclerosis (MS) is a neuroinflammatory disease characterized by chronic focal demyelination and neurodegeneration due to immune-mediated destruction of myelin sheaths and inflammatory attacks towards the neuronal compartment[20,21]. Studies in humans and mice demonstrated a prominent role of CD4 T cells in the pathogenesis of autoimmune demyelinating diseases, in particular of T-bet1/IFNγ-positive T helper 1 (Th1) cells and RORγt/IL-17-positive Th17 cells[22–24]. Since GPCRs play a crucial role in the regulation of T cell trafficking, pharmacological strategies to modulate T cell migration in neuroinflammation have attracted much attention[25].

In this work, we show that P2Y10 facilitates chemokine-induced migration of CD4 T cells through an auto-/paracrine feedback loop involving adenine nucleotides and LysoPS, and that mice with CD4 T cell-specific P2Y10 deficiency show reduced infiltration of CD4 T cells into the spinal cord and reduced disease severity in neuroinflammation.

## Results

**Generation and basal characterization of CD4-P2Y10-KOs.** To identify GPCRs involved in T cell regulation during neuroinflammation, we determined GPCR expression in naïve and spinal cord-infiltrating CD4 T cells during experimental autoimmune encephalomyelitis (EAE), a mouse model of MS. We found that not only chemokine receptors were upregulated, but also various lipid and metabolite GPCRs, very prominent among them *P2ry10*, which encodes P2Y10 (Fig. 1a). Also in circulating CD4 T cells, *P2ry10* was significantly upregulated at peak EAE disease (Suppl. Fig. 1). To investigate the function of P2Y10 in CD4 T cells, we used CRISPR/Cas9 genome editing to flank exon 3 of the X-chromosomal *P2ry10* gene with loxP sites (Fig. 1b) and bred resulting *P2ry10*flox mice to the CD4Cre line, which recombines in all CD4-expressing cells[26]. Female mice with the genotype *CD4*Crepos; *P2ry10*flox/flox and male mice with the genotype *CD4*Crepos; *P2ry10*flox/y (both henceforth CD4-P2Y10-KO) showed a strong reduction of P2Y10 expression in CD4 T cells in immunoblotting (Fig. 1c) and RNA sequencing (Fig. 1d), whereas expression of the related GPCRs *P2ry10b* or *Gpr174*, which are located in close proximity to *P2ry10* on the X chromosome, were not affected (Fig. 1d). We analyzed lymphocyte numbers in various immune organs of adult CD4-P2Y10-KOs, but did not find differences between the genotypes under basal conditions (Fig. 1e–h). Also, the proportion of FoxP3-positive regulatory T cells (Treg) or the proportion of CD44+CD62L− effector memory, CD44+CD62L+ central memory, or CD44−CD62L+ naïve T cells was not altered (Supplementary Fig. 2). We next investigated how the loss of P2Y10 affected CD4 T cell activation, but did not find differences with respect to CD3/CD28-induced cytokine production (Fig. 1i) or proliferation (Fig. 1j). Also after in vitro differentiation towards Th1, Th17, or Treg, we did not observe differences in the proportion of IFNγ-, IL-17- or FoxP3-expressing cells (Fig. 1k) or in the proliferative activity of these in vitro-differentiated cells (Fig. 1l).

**Reduced neuroinflammation in CD4-P2Y10-KOs.** Since P2Y10 is strongly expressed in spinal cord-infiltrating CD4 T cells (Fig. 1a), we investigated EAE development in CD4-P2Y10-KOs. We found that disease severity was significantly lower in CD4-P2Y10-KOs than in control littermates (Fig. 2a), and histological analysis showed reduced demyelination on day 28 (Fig. 2b). To investigate the underlying mechanism, we analyzed leukocyte infiltration at peak disease in a second group of mice (Fig. 2c) and found that numbers of spinal cord-infiltrating CD4 T cells were clearly reduced in KO mice, while CD8 numbers were not significantly altered (Fig. 2d). Within the population of infiltrating CD4 cells, the proportion of FoxP3-expressing Treg or IL-17-expressing Th17 cells was not changed, whereas the proportion of IFNγ-producing Th1 cells was reduced (Fig. 2e). The proportion of GM-CSF expressing CD4 T cells was reduced (Fig. 2f), and this was accompanied by reduced numbers of CD11bpos, Ly6Gneg, CD45hi macrophages in flow cytometric analysis (Fig. 2g) and Mac3-positive macrophages in histological analysis on day 28 (Fig. 2h).

**Reduced migration and polarization in P2Y10-deficient CD4 T cells.** Since activation, proliferation, and differentiation were not altered in P2Y10-deficient CD4 T cells, we next studied their migratory behaviour. Transwell migration assays showed normal basal migration in naïve CD4 (nCD4) cells, but a significant impairment of migration towards homeostatic chemokines such as CCL19, CCL21, and SDF-1α (CXCL12), but also towards inflammatory chemokines such as CCL4, CCL5, CXCL9, or CXCL10 (Fig. 3a). Live cell imaging of CD4 T cells in a homogenous CCL19 field showed that the proportion of immobile cells was increased in P2Y10-deficient cells (Fig. 3b), and when exposed to a CCL19 gradient they travelled shorter distances and displayed reduced migration velocity (Fig. 3c–e). The directness of migration was not significantly changed (Fig. 3f). This impairment of chemokine-induced migration was neither due to a reduced

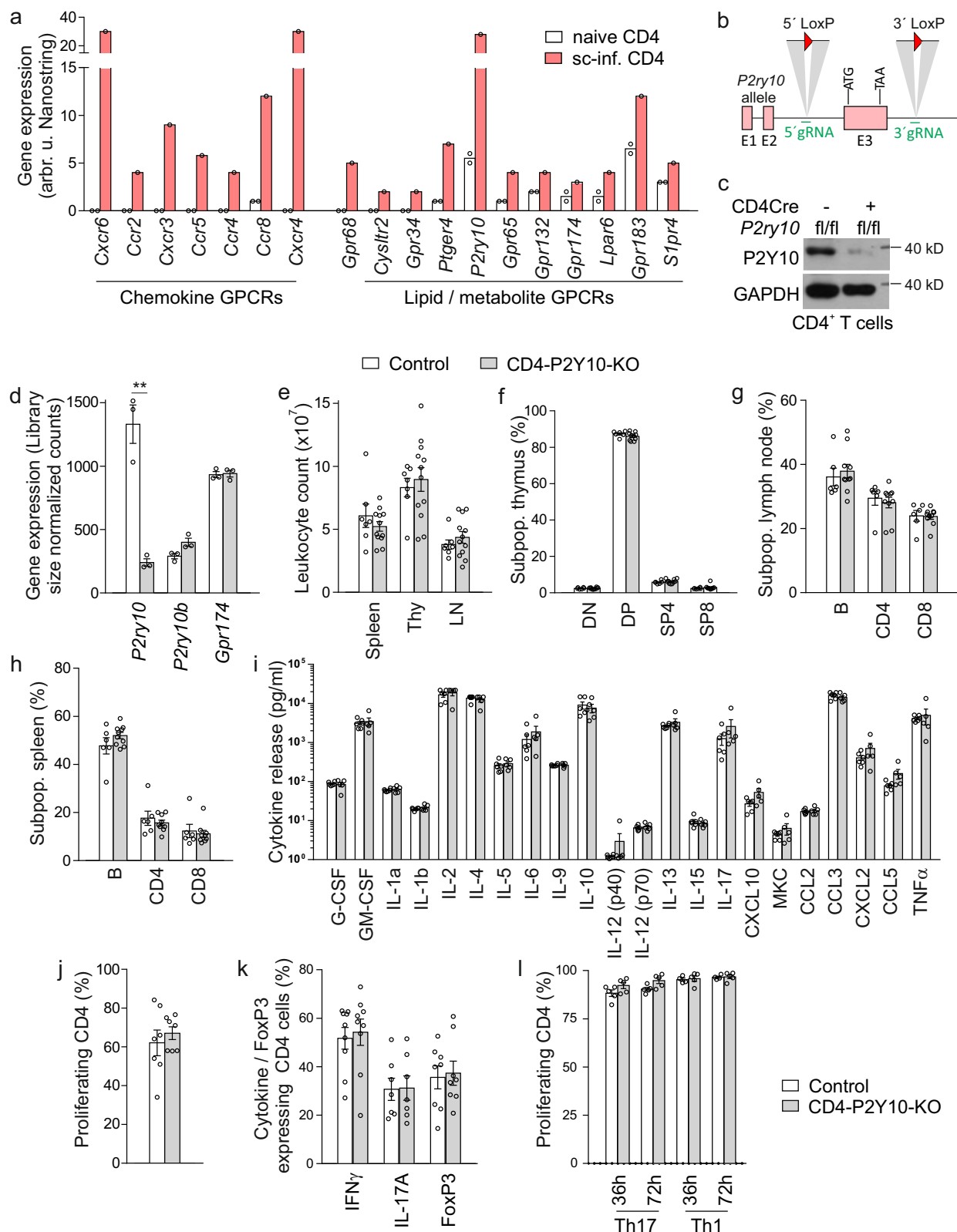

expression of chemokine receptors (Fig. 3g) nor to altered expression of other genes related to T cell migration (selected genes from gene ontology term "T cell migration" (GO:0072678) in Fig. 3h). An increased proportion of immobile cells and reduced migration was also observed in CD3/CD28-activated CD4 T cells

(aCD4) (Supplementary Fig. 3a, i). Also, Th1-differentiated cells showed reduced chemokine-induced migration (Fig. 3j), whereas migration in Th17-differentiated cells was not impaired (Fig. 3k). The latter difference might be due to variations in P2Y10 expression: re-analysis of previously published single-cell

**Fig. 1 Generation and basal characterization of CD4-P2Y10-KOs. a** Expression of selected GPCRs for chemokines or lipids and metabolites in naïve and spinal cord-infiltrating CD4 T cells was determined by Nanostring mRNA quantification (naïve CD4: splenic CD4 T cells from two healthy mice; sc-inf. CD4: spinal cord-infiltrating CD4 T cells pooled from four individual mice at EAE day 14). **b** Design of the floxed P2Y10 allele. **c**, **d** Reduction of P2Y10 expression in CD4 T cells from control and CD4-P2Y10-KO mice was judged by immunoblotting (**c**, GAPDH as loading control) and RNA sequencing in isolated splenic CD4 T cells (**d**, $n = 3$; **$p = 0.0021$). Total leukocyte numbers (**e**) and proportion of lymphocyte populations in thymus (**f**), inguinal lymph nodes (**g**), or spleen (**h**) ($n = 6$–12). Cytokine production (**i**, $n = 5$–6) and proliferation (**j**, $n = 7$) in CD3/CD28-activated CD4 T cells. Percentage of IFNγ/IL-17A/FoxP3-expressing cells (**k**, $n = 7$–9) and proliferating cells (**l**, $n = 4$–5) after Th1 and Th17 differentiation, respectively. Data are means ± SEM (in (**a**) only means); comparisons between genotypes were performed using two-tailed unpaired Student's t-test. $n$, number of mice per group.

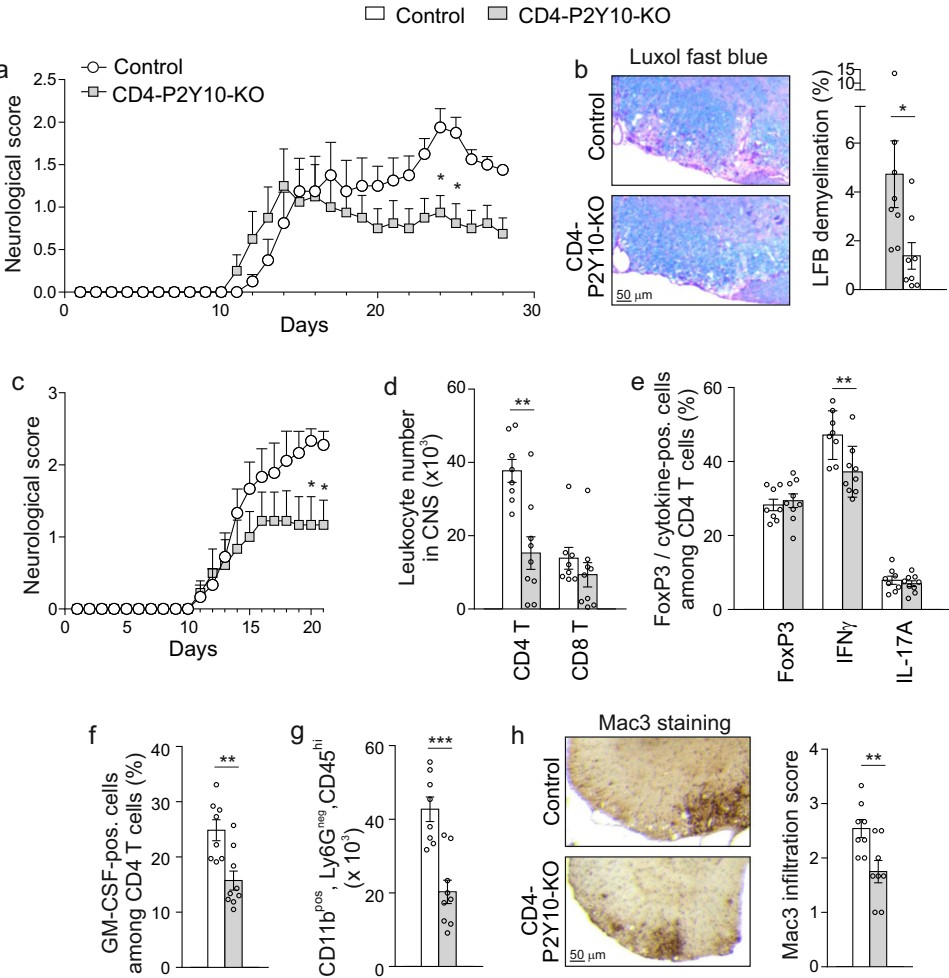

**Fig. 2 Reduced neuroinflammation in CD4-P2Y10-KOs. a** Neurological score ($n = 8$; *$p = 0.037$ (d24), 0.0189 (d25)). **b** Quantification of demyelination by luxol fast blue (LFB) staining on day 28 after induction of EAE: Exemplary photomicrographs (left) and statistical evaluation (right) ($n = 8$; *$p = 0.039$). Repetition of EAE in an independent cohort ($n = 8$–9): Neurological score (**c**; *$p = 0.019$ (d20), 0.033 (d21)) and flow cytometric analysis of spinal cord-infiltrating leukocytes on day 20 (**d**–**g**): Absolute numbers of CNS-infiltrating T cells (**d**; **$p = 0.001$), proportion of FoxP3, IFNγ-, IL-17A- (**e**; **$p = 0.008$) or GMCSF- (**f**; **$p = 0.003$) expressing cells among infiltrating CD4 cells, absolute numbers of CD11b+,CD45hi macrophages (**g**; ***$p < 0.001$). **h** Quantification of Mac3-positive cells in lumbar CNS: Exemplary photomicrographs (left) and statistical evaluation (right) ($n = 8$; **$p = 0.009$) (same mice as in **a**, **b**). Data are means ± SEM; comparisons between genotypes were performed using two-tailed unpaired Student's t-test (**b**, **d**–**h**) or two-way ANOVA with Tukey's multiple comparisons test (**a**, **c**). $n$, number of mice per group.

expression data showed that while 87% of Th1 cells express *P2ry10*, only 37% of Th17 cells are positive for this receptor (Fig. 3l)[10]. To test whether P2Y10 facilitates chemokine-induced migration also in primary human CD4 T cells, we performed siRNA-mediated knockdown in primary CD4 T cells obtained from human blood. Treatment with P2Y10-specific siRNAs resulted compared to control siRNA in a reduction of P2Y10 transcript levels to approximately 40% (Supplementary Fig. 3b). This reduction of P2Y10 levels was sufficient to result in a

significant impairment of SDF-1α-induced migration in CD4 T cells isolated from peripheral blood of healthy donors (Fig. 3m, left). To test whether P2Y10 knockdown had a comparable effect in MS patients, we isolated CD4 T cells from patients that had been diagnosed with MS, but were currently without specific treatment (last corticosteroid application three months before migration analysis). As observed in CD4 T cells from healthy donors, knockdown of P2Y10 resulted in reduced SDF-1α-induced migration (Fig. 3m, right).

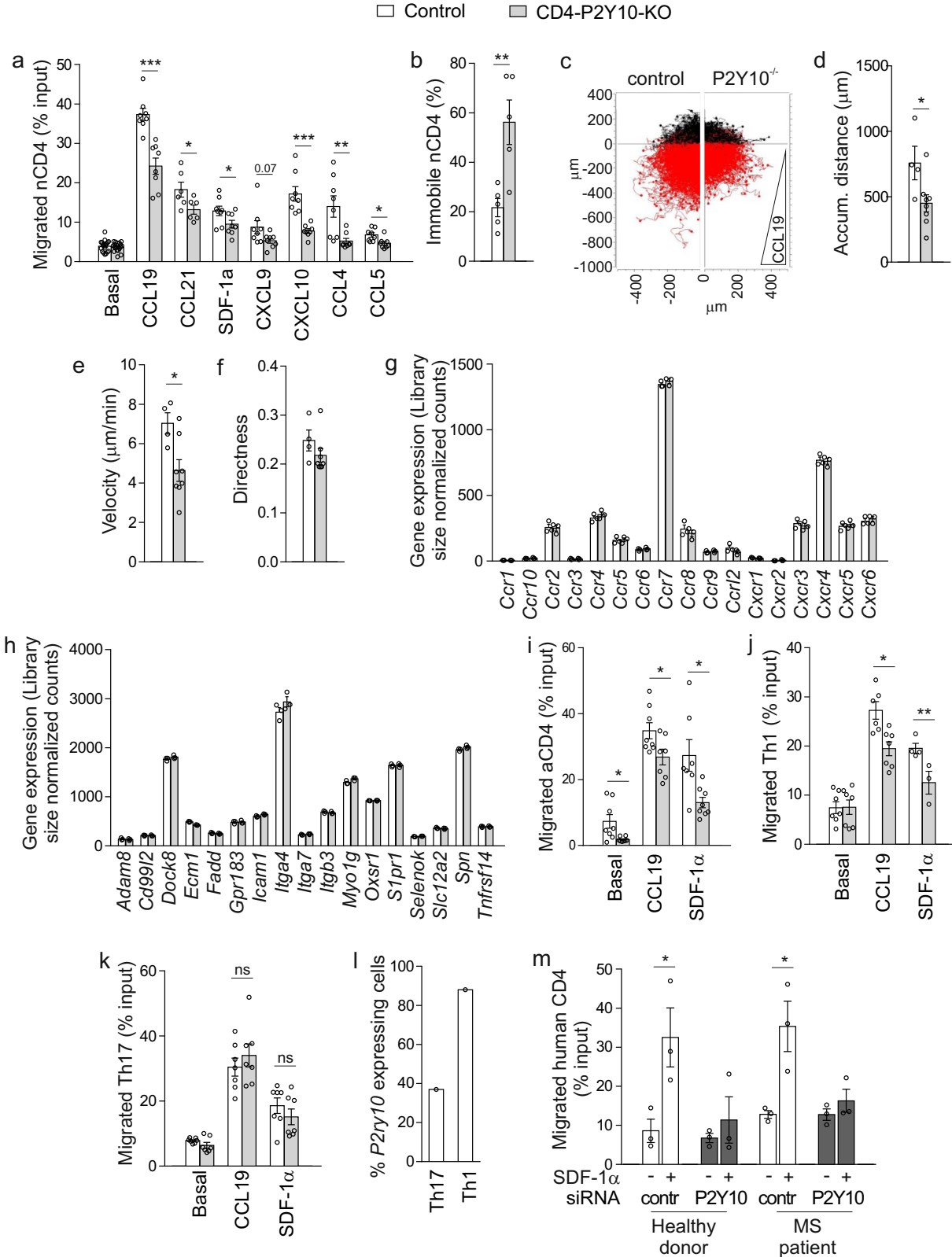

**Reduced chemokine-induced polarization and RhoA activation in P2Y10-deficient CD4 T cells**. To understand the mechanisms underlying impaired migration, we studied chemokine-induced polarization using phalloidin staining. We found that P2Y10-deficient CD4 T cells showed reduced chemokine-induced polarization both in the naïve and CD3/CD28-activated state (Fig. 4a–c). We also assessed the cellular distribution of active

RhoA, a well-known regulator of polarization that is activated both at leading and trailing edge during T cell migration[27–31]. To label active RhoA, we employed the HA-tagged RhoA binding domain of rhotekin, which binds specifically to GTP-bound RhoA, allowing the localization of active RhoA[32]. In CCL19-stimulated control CD4 T cells, active RhoA was found both at rear and front of polarized cells, partially overlapping with

**Fig. 3 Reduced migration and polarization in P2Y10-deficient CD4 T cells. a** Transwell migration of naïve CD4 T cells (nCD4) under basal conditions and after addition of 100 ng/ml of the indicated chemokines to the lower well (6–12 mice per group; $*p = 0.044$ (CCL21), 0.048 (SDF-1α), 0.013 (CCL5); $**p = 0.007$ (CCL4); $***p < 0.001$ (CCL19); $***p < 0.001$ (CXCL10). **b** Live-cell imaging analysis of migratory behaviour in CCL19-stimulated naïve CD4 T cells (homogenous CCL19 application): Quantification of immobile cells (cells from three mice per group were analyzed in five experiments, and per experiment 64–200 cells were evaluated; $**p = 0.007$). Live-cell imaging analysis of migratory behaviour in CCL19-stimulated nCD4 T cells (CCL19 gradient): Exemplary tracking result (**c**) and statistical evaluation of accumulated distance (**d**; $*p = 0.037$), velocity (**e**; $*p = 0.021$), and directness (**f**) (3–5 mice per group were analyzed in 4–8 tracking experiments, and per mouse 937–7969 cells were evaluated). Expression of chemokine receptors (**g**) or other genes related to T cell migration (**h**) in nCD4 (3 mice per group). Transwell migration of CD4 cells that had been unspecifically activated by CD3/CD28 stimulation (aCD4) (**i**, 7–8 experiments with cells from 7 to 8 mice per group; $*p = 0.019$ (bas), 0.038 (CCL19), 0.011 (SDF-1α) or differentiated towards Th1 (**j**, 3–7 experiments with cells from 3 to 4 mice per group; $*p = 0.027$; $**p = 0.005$) or Th17 (**k**, 7 experiments with cells from four mice per group). **l** Percentage of *P2ry10*-expressing cells among Th1- and Th17-differentiated lymph node CD4 T cells (reanalysis of single-cell expression data from ref. [10]). **m** Transwell migration of naïve human CD4 T cells isolated from healthy donors (left) or MS patients (right) after treatment with scramble siRNA or siRNA directed against human P2Y10. Migration was assessed under basal conditions and after addition of 100 ng/ml SDF-1α to the lower well (three patients per group; $*p = 0.025$ (healthy), 0.042 (MS). Data are means ± SEM; comparisons between genotypes were performed using two-tailed unpaired Student's *t*-test. ns not significant.

phalloidin staining (Fig. 4d). In P2Y10-deficient CD4 T cells, the proportion of cells with enrichment of active RhoA at leading or trailing edges was significantly reduced compared to control cells (Fig. 4e, f). For better quantification of chemokine-induced RhoA activation, we determined basal and chemokine-induced activation of small GTPases RhoA using an ELISA-based assay. We found that basal levels of GTP-bound, active RhoA did not differ between naive control and KO CD4 T cells (Fig. 4g), whereas chemokine-induced RhoA activation was abrogated (Fig. 4h). In contrast, levels of active Rac1, another small GTPases centrally involved in the regulation of polarization and migration[27–30], was neither changed under basal conditions nor after chemokine stimulation (Fig. 4i, j). Interestingly, in CD3/CD28-stimulated CD4 T cells RhoA activation is reduced already under basal conditions (Supplementary Fig. 4), coinciding with reduced basal migration (Fig. 3i). This finding might be due to cell-autonomous production of chemokines by these cells (Fig. 1i).

**LysoPS and adenosine triphosphate (ATP) enhance RhoA activity in a P2Y10-dependent manner, and are released in response to chemokine stimulation.** P2Y10 is phylogenetically not related to chemokine receptors, it, therefore, seems unlikely that chemokines are direct activators of P2Y10. However, to exclude any direct effects of chemokines on P2Y10, we investigated CCL19-induced calcium mobilization in COS-1 cells transfected with P2Y10, a promiscuous G-protein, and a calcium sensor protein[33], but did not observe P2Y10-specific responses (Supplementary Fig. 5a). We, therefore, hypothesized that chemokine treatment of CD4 T cells induces the production or release of a mediator that facilitates polarization and migration in a P2Y10/RhoA-dependent manner. Different putative ligands for P2Y10 have been suggested, for example, LPA, S1P, and LysoPS[14,16,17]. In addition, P2Y10 was originally described as a purinergic nucleotide receptor based on sequence homology (see annotation for GenBank file AF000545.1). To test whether any of these suggested ligands induced RhoA activation in a P2Y10-dependent manner, we determined levels of active RhoA 1 min after agonist addition in nCD4 from control and knockout mice. We found that S1P induced a clear increase in RhoA activation in control cells, whereas the effect of LPA was very mild. However, neither S1P- nor LPA-induced RhoA activation was P2Y10 dependent (Fig. 5a). Also, LysoPS species 18:0 and 18:1 enhanced RhoA activation, and this response was abrogated in P2Y10-deficient cells (Fig. 5b). The mild stimulatory effects of other lysophospholipids, such as lysophosphatidylethanolamine (LPE), lysophosphatidylglycerol (LPG), or lysophosphatidylinositol (LPI), did not depend on P2Y10 (Supplementary Fig. 5b). Interestingly, also ATP was able to induce RhoA activation in a P2Y10-dependent manner

(Fig. 5c). To test whether ATP also activates other canonical GPCR effectors through P2Y10 in CD4 T cells, we investigated the production of inositol 1,4,5-trisphosphate (IP3) downstream of $G_{q/11}$ activation (here measured as metabolite inositol monophosphate (IP1) or cAMP levels downstream of $G_s$- or $G_{i/o}$ activation. We found that ATP induced IP1 production in control CD4 cells, but not in P2Y10-deficient CD4 T cells (Fig. 5d). cAMP levels, however, were neither in control nor knockout cells altered by ATP treatment (Fig. 5e, isoprenaline as positive control). In line with a role of P2Y10 as a major ATP receptor in CD4 T cells, we found that *P2ry10* showed the highest expression of all purinergic P2 receptors in CD4 T cells from lymph nodes and spleen (Supplementary Fig. 6a), and reanalysis of published single-cell expression data from murine blood CD4 T cells (GSE 108097) showed that this is also true in circulating CD4 T cells (Supplementary Fig. 6b). Since previous studies did not observe P2Y10-dependent ATP effects in classical ligand screening approaches[34], we also studied ATP-dependent calcium mobilization using P2Y10- or empty vector-transfected COS-1 cells. We found that ATP induced a strong dose-dependent calcium mobilization in empty vector-transfected cells (Fig. 5f), which is due to the endogenous expression of various other purinergic receptors such as P2Y2, P2X2, P2X3, P2Y4 (Supplementary Fig. 6c). However, this response was even further enhanced in P2Y10-transfected cells (Fig. 5f, g), indicating that ATP can indeed induce P2Y10-dependent calcium mobilization in COS-1 cells, even though this effect is easily overlooked due to massive activation of endogenous purinergic receptors. Also, ADP-induced responses were enhanced in P2Y10-expressing cells (Fig. 5g).

We next investigated whether LysoPS and ATP are released from CD4 T cells in response to chemokine stimulation. We found that extracellular ATP concentrations clearly increased 1 and 3 min after the addition of CCL19 (Fig. 5h). Chemokine treatment also induced release of LysoPS 18:0 into the supernatant of CD4 T cells (Fig. 5i), while 18:1 was not detected. The latter might be due to differences in the detection limit of our assay system, which is as low as 5 fmol for LysoPS 18:0, but >100 fmol for LysoPS 18:1. However, chemokine treatment increased both LysoPS species in lysates of CD4 T cells (Fig. 5j, k), and also LysoPS 16:0 levels were increased (Supplementary Fig. 7a). Also, other lysophospholipid species such as lysophosphatidylcholine or lysophosphatidylethanolamine were detected in pellets, but did not change significantly in response to short-term CCL19 stimulation (Supplementary Fig. 7b, c). We also investigated whether lysophospholipid levels changed in spinal cords of EAE mice, and found that LysoPS species 18:0, 18:1, and 22:6, as well as various other lysophospholipids, significantly increased at day 10 after EAE induction (Supplementary Fig. 8). Basal lysophospholipid levels in spinal cords, brains, or sera did not differ between control mice and CD4-P2Y10-KOs (Supplementary Fig. 9).

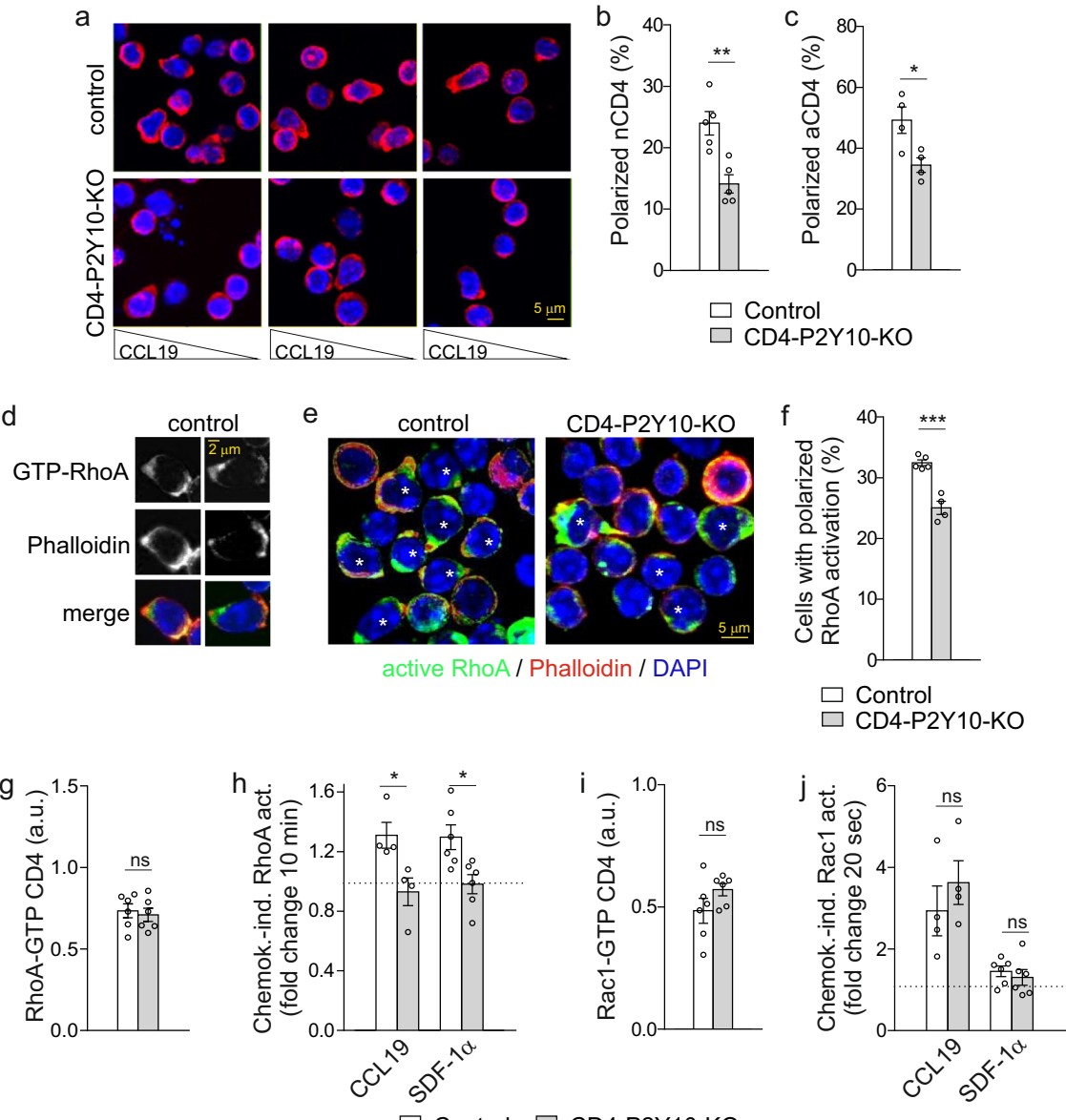

**Fig. 4 Reduced chemokine-induced RhoA activation in P2Y10-deficient CD4 T cells.** Phalloidin staining of nCD4 (**a**, **b**) or aCD4 (**c**) that were allowed to migrate for 10 min towards a CCL19 gradient in µ-Slide Chemotaxis chambers: representative photomicrographs (**a**) and quantification of the percentage of polarized cells in nCD4 (**b**; **, $p = 0.003$) and aCD4 (**c**; *, $p = 0.024$) (4–5 mice per group were analyzed in 4–5 independent experiments, and per experiment 57–259 cells (nCD4) and 32–230 cells (aCD4) were evaluated). **d–f** Distribution of active RhoA in CCL19 (10 min)-stimulated CD4 T cells (homogenous CCL19 environment): **d** Examples showing active RhoA and phalloidin staining in CCL19-polarized control cells; exemplary photomicrographs (**e**) and statistical evaluation (**f**; ***, $p < 0.001$) of cells with polarized RhoA activation (a total of 492 and 302 cells from two mice per group were analyzed (59–124 cells per data point); * indicates cells considered polarized). Basal levels (**g**, **i**) and chemokine-induced changes (**h**, **j**) of active, GTP-bound RhoA (**g**, **h**) and active, GTP-bound Rac1 (**i**, **j**) (4–6 mice per group; **h**, *, $p = 0.024$ (CCL19); *, $p = 0.013$ (SDF-1α)). Data are means ± SEM; comparisons between genotypes were performed using two-tailed unpaired Student's $t$-test. *, $P < 0.05$.

**ATP degradation reduces chemokine-induced RhoA activation and migration in control CD4 cells, but not in P2Y10-deficient CD4 T cells.** To test whether auto-/paracrine mediator release indeed contributed to chemokine-induced T cell migration in a P2Y10-dependent manner, we investigated the effect of apyrase, an ectonucleotidase that degrades ATP and ADP to AMP, on transwell migration. We found that apyrase reduced SDF-1α- and CCL19-induced migration in control CD4 T cells, but not in P2Y10-deficient CD4 T cells (Fig. 6a, b). Also, human CD4 T cells, both from healthy donors and MS patients, showed significantly reduced chemokine-induced migration in the presence

of apyrase (Fig. 6c). Furthermore, apyrase significantly reduced chemokine-induced RhoA activation in murine control CD4 T cells, but not in P2Y10-deficient CD4 T cells (Fig. 6d). We also investigated whether chemokine-induced ATP release and apyrase-mediated inhibition of chemokine-induced migration were observed in CD3/CD28-activated CD4 T cells, and found that this was the case (Fig. 6e, f). These findings suggest that ATP acts as an auto-/paracrine mediator to facilitate chemokine-induced RhoA activation in a P2Y10-dependent manner. Previous studies suggested that chemokine-induced ATP release is mediated by pannexin hemichannels[35], which led us to

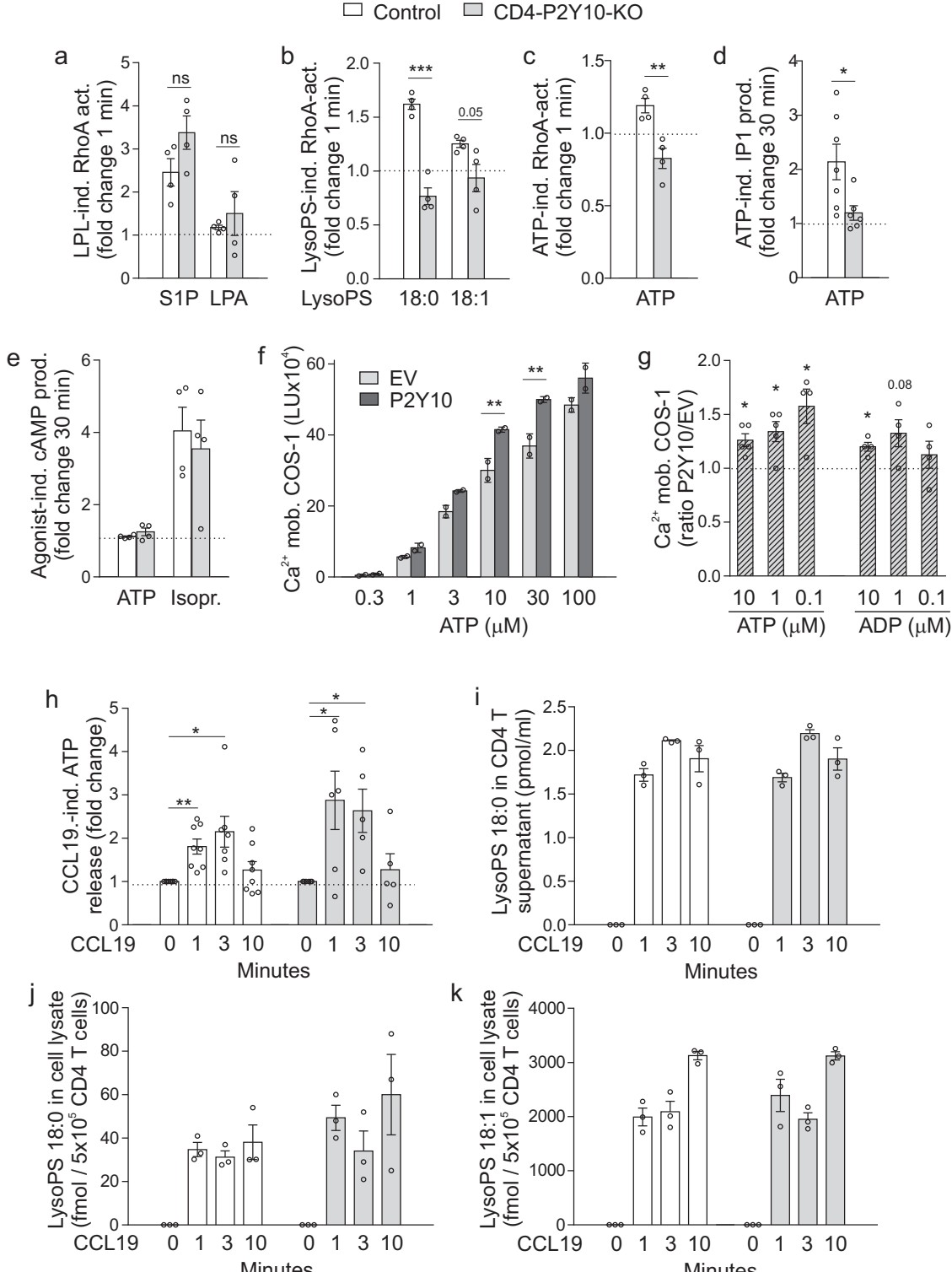

investigate the effect of carbenoxolone, an inhibitor of pannexins and connexins, on chemokine-induced CD4 migration. We found that carbenoxolone significantly reduced CCL19-induced migration in control CD4 T cells, but not in P2Y10-KO cells (Fig. 6g). To further characterize the signalling cascade downstream of P2Y10, we investigated whether heterotrimeric G-proteins of the $G_{12/13}$ family, which are well-known GPCR-dependent activators of RhoA, are required for chemokine-induced migration in CD4 T cells. We studied chemokine-induced migration in cells from mice lacking the α subunits of $G_{12}$ and $G_{13}$, $G\alpha_{12}$ and $G\alpha_{13}$, in

CD4 T cells (CD4-$G\alpha_{12/13}$-KO) and found that they showed significantly reduced migration in response to CCL19 compared to their littermate controls (Fig. 6h).

**P2Y10-dependent facilitation of migration contributes to CD4-mediated responses outside the CNS.** We finally investigated whether the inactivation of P2Y10 would also affect CD4 T cell-mediated diseases outside the CNS. To do so, we used dinitrofluorobenzene (DNFB)-induced contact hypersensitivity, a

**Fig. 5 LysoPS and ATP enhance RhoA activity in a P2Y10-dependent manner, and are released in response to chemokine stimulation.** Changes in the levels of active, GTP-bound RhoA in nCD4 after stimulation with 1 μM S1P or LPA (**a**), 1 μM LysoPS 18:0 and 18:1 (**b**; ***$p < 0.001$), or 10 μM ATP (**c**; **$p = 0.005$) ($n = 4$). **d** ATP (10 μM)-induced changes in IP1 production in nCD4 ($n = 6–7$; *$p = 0.030$). **e** ATP (10 μM)-induced changes in cAMP levels in nCD4 ($n = 4$, isoprenaline (Isopr.) 1 μM as positive control). Calcium mobilization in empty vector (EV)- or P2Y10-transfected COS-1 cells in response to different concentrations of ATP and ADP: **f** original luminescence units (LU) from one exemplary experiment (**$p = 0.007$ (10 μM), 0.003 (30 μM)); **g** ratio of luminescence units in P2Y10- vs. EV-transfected cells in 4–5 independent experiments (*$p = 0.012, 0.022, 0.037 0.016$ (from left to right)). **h** CCL19-induced changes of ATP levels in supernatants of nCD4 ($n = 5–8$; *$p = 0.018$ (contr/3 min), 0.039 (KO/1 min), 0.031 (KO/3 min); **$p = 0.002$ (contr/1 min)). LysoPS species 18:0 (**i**, **j**) and 18:1 (**k**) were determined by LC-MS/MS in the supernatant (**i**) or lysates (**j**, **k**) of CCL19-treated CD4 T cells ($n = 3$). Data are means ± SEM; comparisons between genotypes were performed using two-tailed unpaired Student's $t$-test (**a–e**), two-way ANOVA with Tukey's multiple comparisons test (**f**) or two-tailed one sample T test against 1 (**g**, **h**). $n$, number of mice per group (**a–e**, **h–l**) or independent in vitro experiments (**f**, **g**).

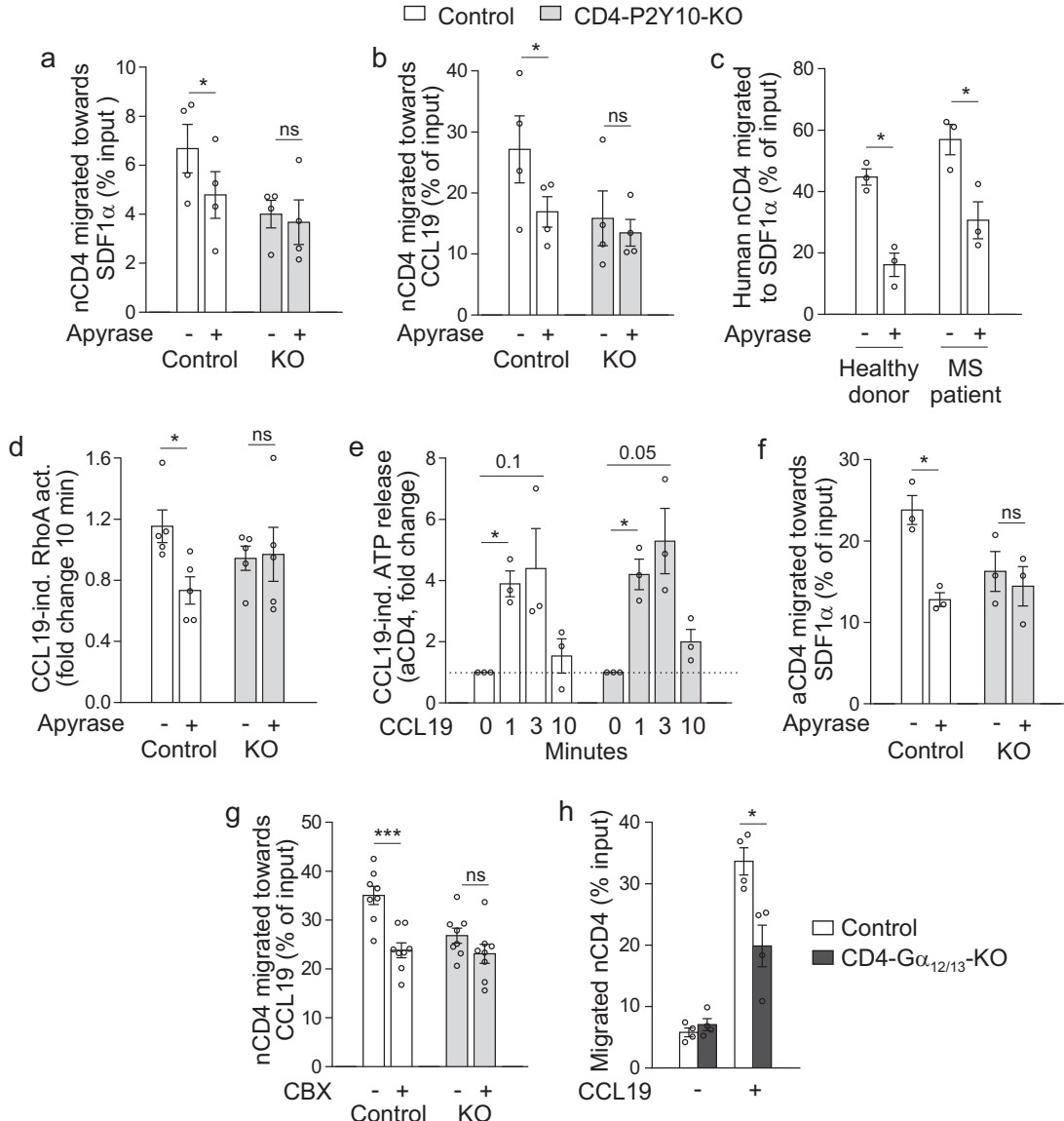

**Fig. 6 ATP degradation reduces chemokine-induced RhoA activation and migration in control CD4 cells, but not P2Y10-deficient CD4 T cells.** Effect of apyrase treatment (20 U/ml) on SDF-1α- (**a**; *, $p = 0.015$) or CCL19- (**b**; *, $p = 0.049$) induced transwell migration in nCD4 ($n = 4$ mice). **c** Effect of apyrase pretreatment on SDF-1α-induced transwell migration in naïve human CD4 T cells isolated from peripheral blood of healthy donors or MS patients ($n = 3$; *, $p = 0.040$ (healthy), 0.034 (MS)). **d** Effect of apyrase pretreatment on CCL19-induced RhoA activation in nCD4 ($n = 5–7$; *, $p = 0.027$). **e** Changes in extracellular ATP concentration after CCL19 stimulation (100 ng/ml) in aCD4 ($n = 3$; *, $p = 0.021$ (contr/1 min), 0.024 (KO/1 min)). **f** Effect of apyrase pretreatment on SDF-1α-induced transwell migration in aCD4 ($n = 3$ mice; *, $p = 0.034$). **g** Effect of carbenoxolone (CBX, 5 μM, 30 min pretreatment) on CCL19-induced transwell migration in murine nCD4 ($n = 8$; ***, $p < 0.001$). **h** CCL19-induced transwell migration in nCD4 from control mice and CD4-Gα$_{12/13}$-KOs (four mice per group; *, $p = 0.014$). Data are means ± SEM; comparisons between groups were performed using two-tailed paired (**a–d**, **f**) or unpaired (**h**) t test or two-tailed one sample t-test versus 1 (**e**). $n$, number of mice or human donors, respectively.

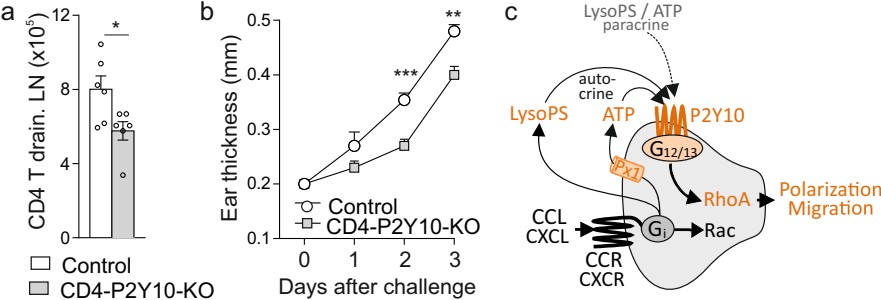

**Fig. 7 P2Y10-dependent facilitation of migration contributes to CD4-mediated responses outside the CNS.** Delayed type hypersensitivity in control mice and CD4-P2Y10-KOs: Flow cytometric analysis of CD4 T cells numbers in draining cervical lymph nodes 48 hrs after challenge (**a**, 6 mice per group; *$p = 0.028$), increase in ear thickness after challenge (**b**, 5 mice per group; **$p = 0.001$; ***$p < 0.001$). **c** Schematic overview of findings: Chemokine stimulation of CD4 T cells results in the release of LysoPS and ATP, which in turn trigger a P2Y10-dependent RhoA activation and consecutive facilitation of polarization and migration. Data are means ± SEM; comparisons between groups were performed using two-tailed unpaired Student's *t*-test (**a**) or two-way ANOVA with Tukey's multiple comparisons test (**b**).

model for responses to foreign antigen[36]. After sensitization to DNFB, mice were challenged by applying the hapten to the skin of the ear, resulting in a Th1-dependent local inflammation with consecutive ear swelling. We found that CD4-P2Y10-KO mice showed decreased numbers of CD4 T cells in draining lymph nodes after challenge (Fig. 7a), and also ear swelling was significantly reduced (Fig. 7b). Together, our data indicate that chemokines induce the release of ATP and LysoPS, resulting in P2Y10/$G_{12/13}$-dependent RhoA activation and consecutive polarization and facilitation of migration (Fig. 7c).

## Discussion

We show in this study that chemokine-induced CD4 T cell migration is facilitated by an auto-/paracrine feedback loop that induces P2Y10-dependent RhoA activation and cell polarization.

Chemokine GPCRs are well-known inducers of leukocyte polarity and locomotion, they establish biochemical asymmetry within leukocytes that results in the formation of an actin-rich front and a myosin II–rich rear[29]. Key players involved in establishing this polarity are phosphoinositide 3-kinases and members of the Rho family of small GTPases, especially Rac, Cdc42, and RhoA[27–30]. Imaging of RhoA activity in transmigrating T cells showed that RhoA is activated at the leading edge, where its activation precedes both extension and retraction events, and in the uropod, where it is associated with ROCK-mediated contraction[31]. RhoA depletion leads to loss of migratory polarity and cells lack both leading-edge and uropod structures, resulting in dysfunctional T cell transendothelial migration[31]. In line with this, T cell-specific RhoA deficient mice show reduced disease severity in autoimmune disorders such as EAE[37]. So far, chemokine receptors were believed to directly mediate RhoA activation during migration[38], but our data indicate that this process involves auto-/paracrine activation of P2Y10.

As mediators of this feedback loop, we identify LysoPS and ATP, which are both released from CD4 T cells upon chemokine stimulation. Immunomodulatory functions of autocrine or paracrine ATP release have been suggested previously, including autocrine facilitation of migration[39,40]. Migrating neutrophils, for example, can release ATP from their leading-edge to amplify chemotactic signals and direct cell orientation by feedback signalling involving the purinergic receptor P2Y2 and the adenosine A3 receptor[41]. In dendritic cells, ATP stimulates P2X7 receptors, resulting in the opening of Panx1 channels and the release of more ATP as part of an autocrine loop that increases DC migration speed[42]. Our data suggest that in CD4 T cells, P2Y10 is a central mediator of ATP-dependent facilitation of migration.

These differences between leukocyte subsets are most likely explained by cell-type-specific patterns of receptor expression: In CD4 T cells, P2Y10 shows the strongest expression of all purinergic receptors, whereas in non-lymphoid cells such as granulocytes the expression of P2Y10 is low compared to other purinergic receptors (Supplementary Fig. 6d, e; data adapted from tabula muris[43]). Like CD4 cells, CD8 cells show high P2Y10 expression both in mice and men. Due to transient Cre expression in the CD4/CD8 double-positive stage[26], also mature CD8 cells show a trend to reduced P2Y10 expression, and chemokine-induced migration was tendentially reduced in these cells (Supplementary Fig. 10). However, this effect was not strong enough to cause a significant reduction of CD8 T cell infiltration into the spinal cord during EAE (see Fig. 2d).

Interestingly, even though the *P2ry10* gene was initially named based on sequence similarity with other P2Y receptors, subsequent phylogenetic analyses produced conflicting results regarding the classification of P2Y10[34]. Using AlignXanalyses, Simon et al. classified P2Y10 as a member of the P2Y family[44], and also two other studies placed P2Y10 within the purinergic receptor family[14,45]. However, others suggested that P2Y10 is more closely related to the platelet-activating factor receptor[13]. Furthermore, analysis of calcium mobilization in CHO cells stably expressing P2Y10 fused to Gα16 failed to detect nucleotide-induced effects, but instead showed responses to LPA and S1P[14]. Based on these inconsistencies, P2Y10 is currently not counted among the established human P2Y receptors, which comprise $P2Y_1$, $P2Y_2$, $P2Y_4$, $P2Y_6$, $P2Y_{11}$, $P2Y_{12}$, $P2Y_{13}$, and $P2Y_{14}$[46]. However, our data suggest that P2Y10 can mediate ATP- and ADP-dependent effects in CD4 T cells, and we also see enhanced calcium responses in P2Y10-transfected COS-1 cells compared to empty vector-transfected cells. However, interpretation of these experiments is certainly hampered by the strong endogenous expression of purinergic receptors in COS-1 (Supplementary. Fig. 6c), which results in massive nucleotide-induced responses even in the absence of P2Y10 expression. This background responsiveness is probably one reason why previous studies failed to observe nucleotide-induced P2Y10 activation.

Regarding the mechanism of chemokine-induced ATP release, our studies employing carbenoxolone point to a role of pannexin channels. Pannexins form large transmembrane channels that allow the passage of ions such as $Ca^{2+}$, but also small molecules such as ATP or inositol triphosphate[47]. Carbenoxolone has broad inhibitory effects on most pannexins and connexins, but since CD4 T cells express on the RNA level predominantly pannexin 1 (Supplementary Fig. 6f), and since on the protein level only

pannexin 1 is detected (Supplementary Fig. 6g, reanalyzed from[48]), we conclude that pannexin 1 is indeed a major contributor to ATP release in these cells. In line with our findings, SDF-1α was shown to induce Panx1-dependent ATP release, polarization, and migration in T cells[35]. Interestingly, as observed in CD4-P2Y10-KOs, Panx1-KOs showed reduced CD4 T cell infiltration in the EAE model[49].

In addition to ATP, also LysoPS is released from chemokine-stimulated CD4 T cells and can mediate P2Y10-dependent RhoA activation in CD4 T cells. This confirms data from a TGFα shedding-based screening approach showing that P2Y10 is activated by LysoPS, and capable of signalling via Gα12/Gα13-containing heterotrimeric G proteins[17]. Based on these findings it is tempting to speculate that both LysoPS and ATP cooperate to facilitate chemokine-induced migration through P2Y10, but due to the lack of suitable pharmacological tools to reduce LysoPS levels, we cannot prove this point experimentally. However, the idea that LysoPS species have immunomodulatory functions is supported by various studies[50]. For example, LysoPS was shown to enhance mast cell degranulation in vitro and in vivo[16,51], to enhance migration of fibroblasts, and to facilitate apoptotic cell-engulfment by macrophages[16,50]. Mutations in the LysoPS lipase ABHD12 cause the human neurodegenerative disorder PHARC (polyneuropathy, hearing loss, ataxia, retinitis pigmentosa, and cataract), and studies in ABHD12-deficient mice suggested that this is due to elevated brain lysoPS levels and consecutively stimulated innate immune cell function[52]. Also, pharmacological inhibition of ABHD12 by DO264 led to elevated LysoPS levels in mouse brain and primary human macrophages and enhanced immunological responses in a mouse model of lymphocytic choriomeningitis virus (LCMV) infection[53]. However, these effects are probably not all mediated by P2Y10, since various other GPCRs have been suggested to respond to LysoPS species. The first LysoPS receptor to be deorphanized was GPR34, an X-linked GPCR that is most abundantly expressed in microglia, capable of coupling to Gαi-containing heterotrimers, and protective in the central nervous system (CNS) against *Cryptococcus neoformans* infection-induced pathology[54,55]. Subsequently, GPR174, P2RY10, and P2RY10b were identified as selective and high-affinity LysoPS receptors using the TGFα shedding screening approach[17]. In support of the functional relevance of LysoPS-mediated GPCR activation, it was shown that LysoPS suppresses the activity and proliferation of murine T cells through the Gs-coupled receptor GPR174[56,57].

Previous screening approaches also suggested S1P and LPA as putative P2Y10 ligands, since they induced intracellular $Ca^{2+}$ mobilization in CHO cells stably expressing P2Y10 fused with a Gα16 protein[14]. These responses were blocked by S1P and LPA receptor antagonists and by RNA silencing of P2Y10, leading to the hypothesis that P2Y10 is a dual S1P/LPA receptor. However, these findings were not confirmed by others[15,34,58]. In line with this, we found that S1P- and LPA-induced RhoA activation was not changed in P2Y10-deficient CD4 T cells.

The extracellular ATP concentration is normally maintained in the nanomolar range, but markedly increases upon tissue damage, inflammation, infection, and other pathological conditions[47,59]. Also lysophospholipid levels rise locally under inflammatory conditions and may serve as danger signals[60,61]. It is therefore tempting to speculate that P2Y10-mediated facilitation of migration is not a purely T cell autonomous mechanism, but can also be driven by local inflammation- or damage-induced increases in the extracellular ATP/LysoPS concentration. However, exogenous addition of ATP or LysoPS did not affect or even inhibited CD4 T cell migration, suggesting that a general increase in ATP/LysoPS levels results in the concomitant activation of other signalling cascades with rather inhibitory potential. In line with this, ATP was shown to reduce in vitro migration of human CD4 T cells to SDF-1α, and this effect was prevented by suramin, indicating a role for P2X receptors[62]. It is well possible that a pro-migratory effect of ATP or LysoPS is only observed when the mediators are acting in an auto- or paracrine fashion, for example, because a certain spatiotemporal resolution relative to the chemokine signal is required.

Taken together, our data show that chemokine stimulation of CD4 T cells results in the release of LysoPS and ATP, which in turn trigger an auto-/paracrine feedback loop resulting in P2Y10-dependent RhoA activation and consecutive facilitation of polarization and migration. In vivo, impaired chemokine-induced migration resulted in reduced infiltration of P2Y10-deficient CD4 T cells into the spinal cord during EAE. It is in this context interesting to note that Th17-differentiated cells are less affected by these changes than Th1-differentiated cells, which is most likely due to differences in P2Y10 expression levels. This differential effect might also underlie the finding that the initial disease phase seems less affected in CD4-P2Y10-KO than the subsequent phases, since Th17 infiltration has been reported to proceed Th1 infiltration during EAE development[63]. Interestingly, even though CCL19-induced migration is known to play an important role in T cell trafficking to lymph nodes under homeostatic conditions[64], we did not detect differences in CD4 T cell numbers in lymph nodes of healthy mice. Of note, CCL19-induced migration is not abrogated, but only reduced in naïve P2Y10-deficient CD4 T cells, indicating that some impairment of CCL19 responsiveness can be tolerated under basal conditions. Under inflammatory conditions, however, loss of P2Y10-dependent facilitation of chemokine-induced migration becomes functionally relevant and results in reduced CD4 T cell infiltration into the target tissue. Therefore, pharmacological inhibition of P2Y10 in CD4 T cells might be a new strategy for the treatment of autoimmune diseases.

## Methods

**Experimental animals**. Mice carrying a floxed *P2ry10* allele (located on the X chromosome) were generated by CRISPR/Cas9 genome editing in murine zygotes as described[65]. In detail, guide 7 (5′-CTGGTAAGTAGGCCTCTTAC-TGG-3, targeting intron 2–3) and guide 5 (5′-CAATGAGTGCACCCTCAATC-AGG-3′, targeting a sequence downstream of exon 3) were designed using https://zlab.bio/guide-design-resources. Homology-directed repair DNAs were designed by flanking the 34 bp loxP sequence (underlined) with 60 bp homology arms corresponding to intronic sequences flanking the PAM site:

*Repair DNA for guide 5*. 5′AGATCTCAGGAGAGTGGGCTCTGTACCTCGCCT GGGCAGCATAGTAGGGCTTGCCCTGATGAATTCATAACTTCGTATAATG TATGCTATACGAAGTTATTGAGGGTGCACTCATTGGGGTGAACCAGCTC TAAGAGCAAGAACACTGGAGAGTAGCCCT 3′ (bold: additional EcoRI site);

*Repair DNA for guide 7*. 5′TGAGATAATTTATATCCTAGAATTTCAGAGAAGC AAGAGAACATGGCACAAAGGCCAGTAGAATTCATAACTTCGTATAATG TATGCTATACGAAGTTATAGAGGCCTACTTACCAGACTATTGATTTGAG TACTAACATAAAATTATCCTGTGGCCTTG-3′ (bold: additional EcoRI site).

Guide RNAs were in vitro-transcribed using the MEGAshortscript Kit (ThermoFischer). A mixture of 50 ng/μl Cas9 mRNA (from TebuBio, Germany), 20 ng/μl of each in vitro-transcribed guide RNA, and 100 ng/μl of each HDR DNA were injected into pronuclei of C57BL6/J zygotes. The first injection resulted in correct insertion of the right loxP site in two male mice, and their offspring were used to target selectively the left side in a second round of zygote injections. Correct integration of both LoxP sites was confirmed in two founder females, and DNA sequencing of potential exonic off-target sites did not reveal relevant off-target effects. For genotyping primer sequences, see Supplementary Table 2.

To generate CD4 T cell-specific P2Y10-deficient mice, we crossed *P2ry10*fl/fl mice to the *CD4*Cre line[26] to obtain *CD4*Cre[pos], *P2ry10*fl/fl (CD4-P2Y10-KO) mice. This term refers both to *CD4*Cre[pos], *P2ry10*fl/fl females and hemizygous males carrying only one of the X-chromosomal *P2ry10*flox alleles (*CD4*Cre[pos], *P2ry10*fl/y). Cre-negative littermates were used as controls (*P2ry10*fl/fl or *P2ry10*fl/y, respectively).

To generate CD4 T cell-specific Gα13/Gα12-deficient mice, *Gna13*fl/fl; *Gna12*−/− mice[66] were crossed to the *CD4*Cre line[26] to generate *CD4*Cre[pos], *Gna13*fl/fl,

$Gna12^{-/-}$ (CD4-G$\alpha_{12/13}$-KO) mice. $Gna13^{fl/fl}$, $Gna12^{+/-}$ littermates were used as controls.

Mice were maintained on a C57BL/6J background and housed under a 12 h light/dark cycle with free access to food and water and under pathogen-free conditions (ambient temperature 20–24 °C, humidity 45–65%). Animal experiments were approved by the Institutional Animal Care and Use Committee of the Regierungspräsidium Darmstadt and in accord with Directive 2010/63/EU of the European Parliament on the protection of animals used for scientific purposes. Mice were analyzed at an age of 7–16 weeks; except for EAE and DTH, both male and female mice were used. Mice were euthanized in deep anesthesia with ketamine 180 mg/kg and xylazine 16 mg/kg (for perfusion and histological analyses) or in $CO_2$ anesthesia and opening of the thorax (if no perfusion was required).

**Animal models.** For active EAE induction, female mice were immunized subcutaneously with 250 µg $MOG_{35-55}$ myelin oligodendrocyte glycoprotein peptide (Genscript) emulsified in Complete Freund's adjuvant (Becton Dickinson) containing 4 mg/ml of heat-inactivated *Mycobacterium tuberculosis* (H37Ra, Becton Dickinson). On days 0 and 2, mice received 500 ng pertussis toxin (Sigma-Aldrich) in PBS intraperitoneally. Clinical scoring of EAE was conducted as follows: 0, no clinical disease; 1, limp tail; 2, impaired righting reflex and gait; 3, partial hind limb paralysis; 4, tetraparesis; 5, dead.

Delayed type hypersensitivity was induced in female mice as described previously[36] and ear thickness was determined daily using a caliper. For flow cytometric quantification of CD4 T cell numbers in draining lymph nodes, cells from three cervical nodes (2 superficial, 1 deep) were pooled per mouse.

**Isolation of murine CD4 T cells.** Untouched $CD4^+$ T cells from lymph nodes or spleen were isolated using the mouse $CD4^+$ T cell isolation kit (Miltenyi, Bergisch-Gladbach, Germany) according to the manufacturer's protocol.

For the analysis of spinal cord infiltrating leukocytes, spinal cords were homogenized in a glass tissue homogenizer in PBS containing 1% glucose and 0.1% BSA. After centrifugation, spinal cords were resuspended in 6 ml of 30% Percoll (Sigma-Aldrich) and layered on a gradient consisting of 4 ml 45% Percoll and 2 ml 70% Percoll. Gradients were spun for 20 min (970 × g, room temperature, without break) and interphases between the layers harvested. After washing, cells were stained with fluorochrome-labelled antibodies and analyzed by flow cytometry.

**Isolation of human PBMC/siRNA-mediated knockdown in primary human CD4 T cells.** Experiments with human samples were performed according to the regulations of the local ethics committee of the Hessian Regional Medical Board (Ethikkommission des Fachbereiches Medizin der Goethe-Universität Frankfurt; AZ 110/11), and informed consent was obtained from all participants. For peripheral blood mononuclear cell (PBMC) isolation, heparinized whole blood was layered over Ficoll (GE Healthcare) and centrifuged at 400×g for 30 min at room temperature (deceleration without brake). The PBMC-containing interphase was washed three times with PBS and analyzed in transwell migration assays. MS patients received the following immunological treatments: 1 patient untreated, 2 patients > 3 months after corticosteroid pulse therapy, 2 patients > 3 months after ocrelizumab treatment, 1 patient on interferon beta-1b therapy. The age of patients was 32–72 years, of healthy donors 32–49 years; Gender was mixed. To knock down the P2RY10 gene in human cells, naïve CD4 T cells isolated from human blood were incubated with Accell P2ry10 siRNA or Accell control siRNA (Horizon a perkin Elmer company) in Accell siRNA delivery medium (Horizon a perkin Elmer company) containing 2 µg/ml anti-hCD3 (clone OKT3, #317301 Biolegend) and 1 µg/ml anti-hCD28 (clone CD28.2, #302933, Biolegend) along with 12.5 ng/mL human rIL-2 (Peprotech) for 72–96 h. Harvested cells were analyzed by quantitative RT-PCR to check the P2y10 knockdown efficiency.

**Induction of effector cells.** CD4 T cells were isolated by magnetic cell sorting (MACS) from naive spleens and peripheral lymph nodes using the $CD4^+$ T Cell Isolation Kit (Miltenyi) according to the manufacturer's instructions.

For effector T cell induction, $1 \times 10^5$ purified CD4 T cells were cultured in 200 µl serum-free X-Vivo 15 medium/well in a U-shaped 96 well plate under the following conditions: For Th17 differentiation, cells were cultured in the presence of 1 µg/ml anti-CD3 (clone 145-2C11 #100339, Biolegend, San Diego, CA, USA), 6 ng/ml anti-CD28 (clone 37.51, #102111, Biolegend) and 10 µg/ml rat anti-mouse IFNγ (clone XMG1.2, #505811, Biolegend) with 2 ng/ml recombinant human TGF-β1 (R&D Systems), 5 ng/ml recombinant murine IL-6 (Promocell), and 20 ng/ml recombinant murine IL-23 (R&D Systems) for 5 days at 37 °C and 5% $CO_2$. For Th1 differentiation, cells were cultured in the presence of 1 µg/ml anti-CD3, 6 ng/ml anti-CD28, and 10 ng/ml recombinant murine IL12 (R&D Systems) for 3 days at 37 °C and 5% $CO_2$. For iTreg differentiation, cells were cultured in the presence of 1 µg/ml anti-CD3, 6 ng/ml anti-CD28 hamster anti-mouse CD28, 5 ng/ml recombinant human TGF-β1 (R&D Systems), and 100 U/ml IL-2 (Peprotech) for 3 days at 37 °C and 5% $CO_2$. After the indicated culture time (see above), Th1 and Th17 cells were activated for 4 h in X-VIVO 15 medium with 50 ng/ml phorbol-12-myristat-13-acetat (PMA, Sigma-Aldrich), 500 ng/ml Ionomycin (Invitrogen), and 1 × Monensin (eBioscience) at 37 °C and 5% $CO_2$. Monensin was added after 1 h

incubation with PMA/Ionomycin. For unspecific CD4 T cell activation, $8 \times 10^5$ cells were cultured in 1 ml serum-free X-Vivo 15 medium containing 20 µl mouse T cell activator CD3/CD28 dynabeads (Life Technologies GmbH, Germany) for 3 days at 37 °C and 5% $CO_2$.

**Flow cytometry.** For the analysis of leukocyte populations from lymphoid organs, the following antibodies were used: rat anti-mouse CD4-FITC (clone GK1.5, #11004182, 1:100 dilution eBioscience), rat anti-mouse CD8a-PE-Cy7 (clone 53-6.7, #25008182, 1:100 dilution,eBioscience) (for thymus); rat anti-mouse CD4-APC (clone RM4-5, #17004281, 1:100 dilution, eBioscience), rat anti-mouse CD45R/B220-FITC (clone RA3-6B2, #554880, Becton Dickinson). For the analysis of spinal cord-infiltrating cells we used rat anti-mouse CD45-PE (clone 30-F11, #12045182, 1:100 dilution, eBioscience), rat anti-mouse Ly6g-APC (clone 1A8, #127613, 1:100 dilution, Biolegend), rat anti-mouse CD11b-eFluor 450 (clone M1/70, #48011282, 1:100 dilution, eBioscience), rat anti-mouse Ly6c-PE-Cy7 (clone HK1.417, #128017, 1:100 dilution, Biolegend), rat anti-mouse F4/80-PerCP-Cy5.5 (clone BM8, #45480182, 1:100 dilution, eBioscience), rat anti-mouse CD62L-FITC (clone MEL14, #553150, 1:100 dilution, BD Bioscience), rat anti-mouse CD44-PE (clone IM7, #553134, 1:100 dilution, BD Bioscience).

For intracellular cytokine staining in spinal cord-infiltrating cells, 500 µl of the sample (= half) were stimulated 4 h in X-VIVO 15 medium with 50 ng/ml PMA (Sigma-Aldrich), 500 ng/ml Ionomycin (Invitrogen), and 1 × Monensin (eBioscience) at 37 °C and 5% $CO_2$. Monensin was added after 1 h incubation with PMA/Ionomycin. After washing, cells were stained following the above-mentioned procedure using antibodies anti IFNγ-PE (clone XMG1.2, #554412, 1:100 dilution, BD Bioscience), anti CD4-APC (clone RM4-5, #17004281, 1:100 dilution, eBioscience), anti-IL17-A-eFluor 450 (clone eBio17B7, #48717782, 1:100 dilution, eBioscience), and rat anti-mouse GM-CSF-PerCP-Cy5.5 (clone MP1-22E9, #505409, 1:100 dilution, Biolegend). For the analysis of spinal cord-infiltrating Treg, the following antibodies were used: anti FoxP3-FITC (clone FJK-16s, #48577382, 1:100 dilution, Thermo Fisher Scientific), rat anti-mouse CD25-Billiant Violet 421 (clone PC61, catalogue number-BLD102033, 1:100 dilution Biolegend), anti CD4-APC (clone RM4-5, #17004281, 1:100 dilution, eBioscience), rat anti-mouse CD152/CTLA-4-PE (clone UC10-4B9, #12152282, Thermo Fischer Scientific), anti CD8a-PECy7(clone 53-6.7, #25008182, 1:100 dilution, Thermo Fischer Scientific), rat anti-mouse TCR β chain-PerCP-Cy5.5 (clone H57-597, #109227, 1:100 dilution, Biolegend). Foxp3 and CTLA-4 were stained intracellular with the buffers of eBioscience, eBioscience™ Foxp3 / Transcription Factor Staining Buffer Set 00-5523-00.

For intracellular cytokine staining in in vitro-differentiated CD4 cells, cells were activated for 4 h X-VIVO™ 15 medium with 50 ng/ml PMA (Sigma-Aldrich), 500 ng/ml Ionomycin (Invitrogen), and 1 × Monensin (eBioscience) at 37 °C and 5% $CO_2$. Intracellular staining was performed using the Foxp3 / Transcription Factor Staining Buffer Set (eBioscience 00-5523-00) according to the manufacturer's protocol. In detail, samples were first incubated with CD16/CD32 (FcBlock, clone 2.4G2, #553141, 1:200 dilution, Becton Dickinson) and anti-CD4-APC for extracellular staining, following intracellular staining using rat anti-mouse FoxP3-FITC (clone FJK-16s, #11577382, 1:100 dilution, Thermo Fisher Scientific) (for iTreg), rat anti-mouse IFNγ –PE (clone XMG1.2, #554412, 1:100 dilution, BD Bioscience) and rat anti-mouse anti-IL17-A (clone eBio17B7, #48717782, 1:100 dilution, eBioscience) (for Th1 and Th17, respectively).

All flow cytometric analyses were performed using a FACS Canto II flow cytometer and FACS DIVA software (version 6.1.2). For gating strategies, please see Supplementary Fig. 11.

**CD4 proliferation.** Cell proliferation was determined using the Cell Trace CFSE Cell Proliferation Kit (Thermo Fisher Scientific) according to the manufacturer's protocol. In detail, $1 \times 10^5$ nCD4 T cells or in vitro-differentiated Th1/Th17 CD4 cells were labelled with 1 µM CFSE for 5 mins at RT in dark. Cells were washed three times and cultured in 200 µl serum-free X-Vivo 15 medium/well of a U-shaped 96 well plate for 3 days in the presence of 1 µg/ml anti-CD3 (clone 145-2C11, #100339, Biolegend). Flow cytometry analysis was performed before the beginning of the culture and after 36 and 72 h of culture (3–5 wells were pooled for each data point). For this, cells were stained with anti-CD4 antibodies and proliferation-dependent dilution of the cell proliferation dye was assessed by flow cytometry.

**Analysis of signalling cascades in primary CD4 T cells.** To determine intracellular cAMP levels, $1 \times 10^6$ CD4 T cells per condition were preincubated with phosphodiesterase inhibitor IBMX (50 µM, 30 min) and stimulated as indicated, then washed in ice-cold PBS, lysed for 10 min with 0.1 M HCl containing 0,1% TritonX100, and centrifuged at 600 × g for 10 min. Supernatants were collected and cAMP levels in the supernatants were measured by using the Direct cAMP ELISA kit (ADI-900-066, Enzo Life Sciences) according to the manufacturer's instructions for the acetylation format.

IP1 production was determined using the IP-One Gq kit following the manufacturer's instructions. In detail, $1 \times 10^5$ CD4 T cells were incubated in the presence of ATP 1 µM for 30 min, and luminescence was determined using luminometric plate reader (Flexstation 3; Molecular Devices).

Active, GTP-bound RhoA or Rac1 were determined with RhoA or Rac1 G-LISA activation kits (Cytoskeleton, Denver, CO, USA). For each measurement, $2 \times 10^6$ CD4 T cells were used; basal levels and agonist-induced changes in activation were determined at the indicated times after addition of CCL19 (100 ng/ml, Biolegend), SDF-1α (100 ng/ml, Biolegend), LPA (1 μM, Sigma-Aldrich), S1P (1 μM, Cayman Chemicals), ATP (10 μM, ThermoFisher), LysoPS 18:0, LysoPS 18:1, LPG, LPE, LPI (1 μM, Avanti Polar Lipids). In some cases, apyrase (20 units/ml, Sigma-Aldrich) was added together with agonists.

**Transwell migration assays.** Isolated CD4 T cells or human PBMC cells were suspended at a density of $1 \times 10^6$ cells/ml in RPMI1640 + 10 mM HEPES + 0.1% BSA. After 1 h of serum-starvation, 50 μl of the cell suspension were added to 5 μm pore size 96 well inserts (Corning, Acton, MA, USA). The lower wells contained either 250 μl medium alone (RPMI1640 with 10 mM HEPES and 0.1% BSA) or medium plus CCL19, CCL21, CCL4, SDF-1α, CXCL9, CXCL10, and CCL5 (all 100 ng/ml, Peprotech). In some cases, apyrase grade VII (20 units/ml, Sigma-Aldrich) or carbenoxolon (5 μM, 30 min pretreatment, Cayman Chemicals) was added together with agonists. Cells were allowed to transmigrate for 3 h at 37 °C and 5% $CO_2$, then transwell plates were kept on ice for 20 min and centrifuged at $180 \times g$ for 3 min. Inserts were discarded, and the number of cells transmigrated compared to input was determined by flow cytometry. "% input" was calculated as (number of transmigrated CD4 T cells/number of CD4 T cells added to the upper well) × 100.

**Live cell imaging.** For live-cell imaging of undirected CCL19-induced migration, $1 \times 10^5$ naïve CD4 T cells were seeded in RPMI medium (RPMI with 20 mM HEPES, 0.1%FBS) in a 96 well plate. After 1 h of serum starvation at 37 °C and 5% $CO_2$, cells were stimulated with CCL19 (100 ng/ml) and phase-contrast images were captured using the Olympus IX81 live cell Imaging system (10/0.3 objective lens, 7 frames per min). For the determination of the proportion of immobile cells, individual cells were manually tracked during a 20 min time-lapse movie and classified as follows: (1) immobile, not polarized, (2) stationary but polarizing, (3) polarizing and migrating, (4) only passive movement (floating). The proportion of immobile, not polarized cells (category 1) is expressed as the percentage of all analyzed cells within the same view field.

For live-cell imaging of directed CCL19-induced migration, 3D chemotaxis assays were performed using μ-Slide Chemotaxis chambers (Ibidi) following the manufacturer's instructions application note 23. In brief, $1 \times 10^8$ CD4 T cells/ml were suspended in a neutralized collagen solution ($1 \times$ DMEM containing 1.6 mg/ml collagen I (CORNING, USA), 1 % FBS and 0.3% NaHCO3). 6 μl of the collagen gel suspension was seeded into the narrow channel of the (IbiTreat) μ-Slide Chemotaxis chamber and allowed to polymerize at 37 °C and 5% $CO_2$. After 1 h, the chemotaxis chamber was filled with bicarbonate-free RPMI 1640 medium containing 10 mm HEPES, which was supplemented with 1% heat-inactivated FBS and antibiotics. Next, RPMI containing CCL19 (100 ng/ml) and 1%FBS was drawn into one of the reservoirs, and RPMI containing 1% FBS was added to the opposite reservoir. Chambers were incubated for 30 min to allow for the establishment of chemokine gradient. Phase-contrast images were captured using the Olympus IX81 live cell imaging system (4× objective lens) equipped with an automated stage and a 37 °C environmental chamber. Images were collected at 1 min interval for 4 h. Imarisx64 9.7.2 was used for the analysis and automated tracking of cells (for an example, see Supplementary Movie 1). The accuracy of automated tracking was manually controlled and the analysis was restricted to motile cells by eliminating cells that are not moving for more than 60 s. The average cell velocity, directness, and accumulated distance travelled were evaluated using the ibidi Chemotaxis and Migration Tool, according to the manufacturer's instructions.

**Determination of polarization.** Polarization and morphological characteristics of the CD4 T cells were investigated using phalloidin staining according to ibidi chemotaxis staining application note 44. Confocal imaging of stained cells was performed using a Leica TCS SP5 II and analyzed using ImageJ (National Institutes of Health). The polarization status of individual CD4 T cells was judged based on length/width ratio and distribution of phalloidin-stained actin.

To analyze the intracellular distribution of active RhoA, $1 \times 10^6$ naïve CD4 T cells were seeded on chambered coverslips coated with 0.01% collagen and cells were stimulated with 100 ng/ml murine CCL19 for 5 min. Cells were fixed with 4% formaldehyde and permeabilized with 0.1% triton X-100 for 4 min and blocked with 1% BSA in TBS for 1 h at room temperature. Cells were incubated with the GST-tagged Rho binding domain (RBD) of human rhotekin (#RT01, 1:100 dilution, Cytoskeleton) for 24 h at 4 °C, washed twice in TBST, then incubated with goat anti-GST antibodies (#27457701, 1:200, Merck) overnight at 4 °C. After washing twice in TBST, cells were incubated with FITC-labelled donkey anti-goat secondary antibodies (#A11055, 1:500, Thermo Fischer Scientific) for 1 h at room temperature, followed by staining with Alexa Fluor™ 594-labelled phalloidin (#A12381,1:100, Thermo Fischer Scientific) and DAPI staining. Confocal imaging of stained cells was performed using a Leica TCS SP5 II. The proportion of cells with polarized RhoA activation (either rear or front or both) was determined manually and data are expressed as the percentage of all analyzed cells within the same view field.

**Calcium mobilization in transfected COS-1 cells.** COS-1 cells, obtained from ATCC, were seeded in 96-well plates with white walls and transparent bottom and transfected with plasmids containing cDNAs encoding a calcium-sensitive biolu-minescent fusion protein consisting of aequorin and GFP[33] and the indicated receptors by using Lipofectamine 2000 (Life Technologies) following the manufacturer's instructions. Forty-eight hours later, cells were loaded with 5 μM coelenterazine h (Promega) in HBSS containing 1.8 mM calcium and 10 mM glucose for 2 h at 37 °C. Calcium transients were measured for 2 min after agonist application using a luminometric plate reader (Flexstation 3; Molecular Devices). The area under each calcium transient was calculated by using SoftMaxPro software and expressed as area under the curve (AUC).

**Multiplex bead array assay.** Cytokines and chemokines in the supernatant of CD3/CD28-activated CD4 T cells (3 days of stimulation) were determined according to the manufacturer's protocol using a mouse cytokine/chemokine multiplex bead array (Millipore, Millerica, MA, USA) and a Luminex MAGPIX analyzer.

**ATP production.** Extracellular ATP levels were determined using the ATP Determination Kit (Invitrogen) according to the manufacturer's instructions. In brief, $5 \times 10^5$ cells per condition (in 150 μl RPMI) were pretreated with 30 μM ectonucleotidase inhibitor ARL68156 for 20 min, followed by agonist stimulation as indicated.

**Western blotting.** Samples were lysed in RIPA buffer (RIPA Lysis and Extraction Buffer, cat. 89900, Thermo Fisher Scientific) (25 mM Tris-HCl pH 7.6, 150 mM NaCl, 1% NP-40, 1% sodium deoxycholate, 0.1% SDS) supplemented with protease inhibitor cocktail (cOmplete™ ULTRA Tablets, Roche). Proteins were separated by SDS-PAGE (Tris-glycine gels with Tris/glycine/SDS buffer) and transferred onto nitrocellulose membranes (Whatman, Dassel, Germany), using the Mini Trans-Blot® Cell (Bio-Rad). Membranes were probed over night with antibodies directed against P2Y10 (Life Technologies GmbH, #PA570914, 1:1000) or GAPDH (Cell Signalling Technology, #2118, 1:1000). After washing, membranes were probed with horseradish peroxidase-conjugated antibodies directed against rabbit or mouse IgG (1:3000, Cell Signalling Technology Europe). The target proteins were visualized by enhanced chemiluminescence reagent (Millipore, Billerica, MA, USA) and a ChemiDoc MP Imaging System (Bio-Rad).

**Histological analyses.** Mice were euthanized under deep anesthesia (ketamine 180 mg/kg; xylazine 16 mg/kg) by intracardiac perfusion with phosphate-buffered saline (PBS) followed by perfusion with 4% (w/v) paraformaldehyde (PFA) solved in PBS. Brain and spinal cord were removed and fixed in 4% PFA overnight. Prior to paraffin embedding, the spinal cord was cut into seven to ten transverse segments (3 mm thick) and coronal brain cuts were made. Sections (3 μm) were stained by HE (hematoxylin and eosin) and LFB-PAS (luxol fast blue including periodic acid-Schiff). Immunohistochemistry was performed using a biotin-streptavidin peroxidase technique (Dako) and an automated immunostainer (AutostainerLink 48, Dako). Sections were pretreated with citrate buffer (pH 6) in a steamer for immunohistochemistry for Mac3 (clone M3/84, #550292, 1:200 also known as CD107b or LAMP-2; BD Pharmingen). Quantification of demyelination and Mac3 infiltration was performed as described previously[67,68].

**General statistical analyses.** Data are presented as means ± standard errors of the means (SEM). Statistical analyses of two groups were performed using two-tailed unpaired t-test (comparison of one variable between two genotypes) or paired t-test (comparison of the same sample under two conditions, e.g., drug treatment). Comparisons between more than two groups or two groups at different times or treatments were analyzed by one way ANOVA or two way ANOVA as indicated. Normalized data (control group set to 1) were analyzed by one-sample t-test. Normality was tested using the Kolmogorov−Smirnov test. $P$ values are indicated as follows: $*p < 0.05$; $**p < 0.01$; $***p < 0.001$.

**Quantification of lysophospholipids by LC-MS/MS.**

1. Analysis of LysoPS species in cell pellets and supernatants.
   Cell culture sample pellets or 50 μL cell culture supernatant were mixed with 300–350 μL sample preparation buffer (30 mM citric acid, 40 mM disodium hydrogen phosphate, pH 4). In some cases, 4 μl of the labelled lipid 18:1-d7 Lyso-PC (500 ng/mL) were added to the sample as internal standard. An acidic pH was used here as it was shown to neutralize the charge of acidic lipids like LPA and LPS and therefore increase their recovery from the organic phase[69,70]. Lipids were extracted the following way: 750 μL 1-butanol was added, samples were vortexed, 350 μL water-saturated 1-butanol was added and the samples were vortex again. After a 1 min sonication step, the samples were centrifuged for 1 min at $500 \times g$ and the resulting organic phase was transferred to a new vial. The 1-butanol extraction was performed again. Both organic phases were pooled and evaporated to dryness in a vacuum centrifuge (Concentrator plus, Eppendorf, Hamburg, Germany). Each sample was resuspended in 5 μL 1-butanol and diluted with 20 μL LC mobile phases (50% A, 50% B, in some

cases 7.5 µL 1-butanol and diluted with 30 µL LC mobile phases were used). Liquid chromatography mobile phases were composed of A: 5:1:4 isopropanol/methanol/water containing 0.2% formic acid, 0.15% ammonium hydroxide, and +1 ml/L InfinityLab deactivator additive (methylendiphosphonic acid from Agilent, Waldbronn, Germany), B: 9:1 isopropanol/water containing 0.2% formic acid, 0.15% ammonium hydroxide, and 1 ml/L InfinityLab deactivator additive. Samples were vortexed and centrifuged again, and clean supernatants were transferred into glass vials ready for ESI-LC MS/MS measurement.

Positive electrospray ionization LC-MS/MS was performed on an Agilent 1290 Infinity LC system (Agilent, Waldbronn, Germany) coupled to a QTrap 5500 mass spectrometer (Sciex, Darmstadt, Germany). Ion source parameters were the following: CUR 25 psi, CAD medium, Ion Spray 4500 V, TEM 300 °C, GS1 30 psi, GS2 40 psi. The following Q1/Q3 transitions were included in this targeted Multiple Reaction Monitoring (MRM) method: 498.3/313.2 for PS(16:0/0:0), 526.2/57.1 for PS(18:0/0:0), 524.2/339.3 for PS(18:1/0:0), 762.5/577.5 for PS(16:0/18:1), 790.5/605.5 for PS(18:0/18:1), and 792.6/607.5 for PS(18:0/18:0). In a second set of experiments, where a polarity switching method was applied, the following parameters were set: CUR 35 psi, CAD medium, Ion Spray Voltage −/+4500 V (for negative and positive mode), TEM 300 °C, GS1 45 psi and GS2 45 psi. Here, 12 LPS lipids were included in the targeted MRM screen (details of the targets are outlined in Suplementary Table 1).

Reversed-phase LC separation was performed using a Kinetex C18 column (1.7 u, 100 × 2.1 mm). Before starting the run, the LC-system was washed with 0.5% phosphoric acid for 15 min. Compounds were eluted with a gradient starting with 30% B, increasing in 4 min from 30 to 50% B, in 2 min from 50 to 80% B, in 1.5 min from 80 to 95% B, holding 95% for 1.5 min, decreasing in 0.5 min from 95 to 5% B, hold for 1 min, increase in 0.5 min to 30% B and hold for 2 min. Column oven temperature was set to 40 °C. Autosampler was set to 8 °C (20 °C in the 2nd set). The injection volume was 5 or 7.5 µL. Calibration curves were performed with authentic standards. Analyst 1.6.2 and MultiQuant 3.0 (both from Sciex, Darmstadt, Germany), were used for data acquisition and analysis, respectively.

2. General analysis of lysophospholipids in sera, tissues, cell pellets, and supernatants (global lipid extraction).

*Extraction of lipids from serum.* 10 µl of serum from each sample were diluted with 100 µl of 1-butanol/methanol (1:1; v/v; containing 5 mM ammonium formate). In addition, 4 µL of the labelled lipid 18:1-d7 Lyso-PC (500 ng/mL) were added to each sample as internal standard. Samples were vortexed for 10 sec, followed by a sonication step for 10–15 min in a water bath at 20 °C. Next, samples were centrifuged at 16,000 × g for 10 min at 20 °C. The generated supernatants were transferred to MS glass vials ready to be analyzed by the mass spectrometer.

*Extraction of lipids from tissues.* Tissues were mashed in a BioMasher and the weight of the homogenized tissues was calculated. The 10 × volume of 1-butanol/methanol (1:1; v/v; containing 5 mM ammonium formate) was added to each tissue. The internal standard was added as above. Samples were vortexed for 10 s, followed by a sonication step for 60 min in a water bath at 20 °C. During the 60 min, samples were shortly vortexed frequently. When sample homogenization was reached, they were centrifuged at 16,000 × g for 10 min at 20 °C. The generated supernatants were transferred to MS glass vials ready to be analyzed.

*Extraction of lipids from cell pellets.* A pellet containing $5 \times 10^5$ cells was diluted in 50 µl of 1-butanol/methanol (1:1; v/v; containing 5 mM ammonium formate). The extraction was done in the same way as for serum.

*Extraction of lipids from medium supernatants.* 100 µl of supernatant was diluted in 400 µl 1-butanol/methanol (1:1; v/v; containing 5 mM ammonium formate). The internal standard was added as above. Samples were vortexed for 10 s, sonicated for 5 min in a water bath at 20 °C, and centrifuged at 16,000 × g for 10 min at 20 °C. The generated clean supernatants were transferred to MS glass vials ready to be analyzed.

Polarity switching electrospray ionization LC-MS/MS was performed on an Agilent 1290 Infinity LC system (Agilent, Waldbronn, Germany) coupled to a QTrap 5500 mass spectrometer (Sciex, Darmstadt, Germany). Ion source parameters were as follows: CUR 35 psi, CAD medium, Ion Spray Voltage −/+ 4500 V (for negative and positive ionization modes), TEM 300 °C, GS1 50 psi, GS2 50 psi. In total, 74 lyso-phospholipids were included in this targeted MRM screen, represented by 13 Lyso-PC (LPC) lipid species, 12 Lyso-PI (LPI), 12 Lyso-PG (LPG), 12 Lyso-PE (LPE), 12 Lyso-PS (LPS) and 13 Lyso-PA (LPA) species. Supplementary Table 1 summarizes the lipids, including information about Q1/Q3 transitions, dwell time, declustering potential (DP), collision energy (CE) and collision cell exit potential (CXP). LPC, LPI, LPG, and LPE were screened in positive ionization mode, LPA in negative ionization mode and LPS in both modes (polarity switching). Reversed-phase LC separation was performed by using a Zorbax Eclipse Plus C18 RR HD column (2.1 × 50 mm; 1.8 µm) with the Zorbax Extend-C18 2.1 as a guard. Compounds were eluted with a flow rate of 0.3 ml/min and with the following 30 min gradient: 0–1.5 min 68% A, 1.5–4 min 55% A, 4–5 min 48% A, 5–8 min 42% A, 8–11 min 34% A, 11–14 min 30% A, 14–18 min 25% A, 18–21 min 3% A, 21–25 min 3% A, 25–25.1 min 68% A, and 25.1–30 min 68% A. Solvent A consisted of AcN/water (60:40, v/v) with 10 mM ammonium formate and solvent B consisted of IPA/AcN (90:10, v/v) with 10 mM ammonium formate. The column oven temperature was set to 30 °C, and the Autosampler was set to 20 °C. Injection volume was 5 µL. For the quantification of the lyso-

phospholipids (concentration in nmol and/or pmol/mg and/or nmol/L), linear calibration curves were performed with the following authentic standards: LPS 18:1 used for all LPS, LPC 18:0 for all LPC, LPI 18:0 for all LPI, LPG 18:0 for all LPG and LPE 18:0 for all LPE. Analyst 1.6.2 and MultiQuant 3.0 (both from Sciex, Darmstadt, Germany), were used for data acquisition and analysis, respectively.

*Cross-validation of lysophospholipids to ensure correct identification*: To ensure correct identification of the lysophospholipids, especially for the ones with a missing standard, all authentic standards and 1–2 samples per biomaterial (serum, tissues, etc.) were cross-validated in a non-targeted way on an Orbitrap QExactive System (Thermo Fisher Scientific) by using the same LC conditions as mentioned above in the two methods. The accurate mass and fragmentation pattern of the lyso-phospholipids was used for identification, their retention time, and elution order was used for cross-validating the QTrap output with the QExactive output.

**Transcriptional analyses.** For RNA sequencing, RNA was isolated from MACS-sorted CD4 T cells using the RNeasy micro kit. (Qiagen) combined with on-column DNase digestion (DNase-Free DNase Set, Qiagen) to avoid contamination by genomic DNA. RNA and library preparation integrity were verified with LabChip Gx Touch 24 (Perkin Elmer). 500 ng total RNA was used as input for VAHTS Universal V6 RNA-seq Library Prep Kit following the manufacture's protocol for stranded mRNA libraries (Vazyme). Sequencing was performed on the NextSeq500 instrument (Illumina) using v2 chemistry, resulting in average of 31 M reads per library with 1 × 75 bp single end setup. The resulting raw reads were assessed for quality, adapter content, and duplication rates with FastQC version 0.11.8 (Available online at: http://www.bioinformatics.babraham.ac.uk/projects/fastqc). Trimmomatic version 0.39 was employed to trim reads after a quality drop below a mean of Q20 in a window of five nucleotides[71]. Only reads between 30 and 150 nucleotides were cleared for further analyses. Trimmed and filtered reads were aligned versus the Ensembl mouse genome version mm10 (GRCm38) using STAR 2.7.3a with the parameter "–outFilterMismatchNoverLmax 0.1" to increase the maximum ratio of mismatches to mapped length to 10% (Dobin et al., STAR: ultrafast universal RNA-seq aligner). The number of reads aligning to genes was counted with featureCounts 1.6.5 tool from the Subread package[72]. Only reads mapping at least partially inside exons were admitted and aggregated per gene. Reads overlapping multiple genes or aligning to multiple regions were excluded. The Ensemble annotation was enriched with UniProt data (release 06.06.2014) based on Ensembl gene identifiers (Activities at the Universal Protein Resource (UniProt)). The raw count matrix was batch corrected using CountClust[73] and then normalized with DESeq2. Differentially expressed genes were identified using DESeq2 version 1.26.0 (Love et al., Moderated estimation of fold change and dispersion for RNA-Seq data with DESeq2). Data are displayed as library-size normalized counts.

NanoString analysis of GPCR expression in bulk RNA was performed as described previously[74]. In brief, 250–500 ng RNA from sorted cells was applied in a total volume of 30 µl in the assay. Barcodes were counted for ~1,150 fields of view per sample. Counts were first normalized to the geometric mean of the positive control spike counts, then a background correction was done by subtracting the mean + two standard deviations of the eight negative control counts for each lane. Data were then normalized to the geometric mean of reference genes Hprt, Rpl22, and Tbp.

**Expression analysis by qRT-PCR.** RNA isolation and reverse transcription were performed using the Quick RNA Micro kit (Zymo, Freiburg, Germany) and ProtoScriptII (NEB, Ipswich, MA, USA) according to the manufacturer's instruction.

Gene expression levels were measured by real-time PCR using SYBR green PCR mix (Roche) using a Light Cycler 480 II (Roche). Gene expression was normalized to the endogenous control (*Gapdh*) and calculated using the ΔΔCt method. For primer sequences, see Supplementary Table 2.

**Reporting summary.** Further information on research design is available in the Nature Research Reporting Summary linked to this article.

## Data availability

The mRNA sequencing data have been deposited in the GEO database under accession code https://www.ncbi.nlm.nih.gov/geo/query/acc.cgi?acc=GSE162246. Source data are provided with this paper.

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

## Acknowledgements
We thank Martina Finkbeiner, Ulrike Krüger and Claudia Ullmann for their expert technical assistance. This project was supported by the Deutsche Forschungsgemeinschaft (DFG) Grant CRC128 Project A03 (to NW) and Z02 (to TK), as well as CRC1039 Project A06 (to IF), A10 (to NW), B08 (to RK and WP).

## Author contributions
M.G. performed most in vitro studies and discussed data; D.T. generated CD4-P2Y10-KOs, performed most in vivo analyses, and discussed data; S.K. and S.Z. performed mass spectrometric analyses, J.C.S. and R.B. helped with in vitro studies; S.G. performed transcriptional analyses; T.K. provided histological analyses; R.K. and K.S. provided patient material, I.F. and S.O. discussed data and manuscript; N.W. initiated and supervised the study, analyzed and discussed data and wrote the manuscript.

## Funding

## Competing interests
The authors declare no competing interests.
