## [Peer Review File · Nature Communications]

G-protein-coupled receptor P2Y10 facilitates chemokine-induced CD4 T cell migration through autocrine/paracrine mediatorsREVIEWER COMMENTS

Reviewer #1 (Remarks to the Author):

Provided is a careful investigation of P2Y10 in CD4+ T cells describing its expression, role in cell migration, (autocrine) mechanism of action, and effect in the EAE model of neuroinflammation. Experiments are novel, informative, well conducted, and clearly described.

I have a few comments:

1. The link with MS provided in the Introduction reads somewhat exaggerated and is not necessary for understanding the context or appreciating the value of this study. I suggest shortening this part or providing data that indeed relate to MS.
2. Fig 1A compares GPCR expression between naïve and spinal cord-infiltrating CD4+ T cells. Has GPCR expression also been measured in mature, circulating CD4+ T cells? And how about CD8+ T cells? In our own, unpublished data of human brain-resident and paired circulating T cells, we notice that the P2RY10 gene is expressed more abundantly than other P2X/P2Y receptor gene and at comparable levels in CD4+ vs CD8+ as well as brain- vs blood-derived T cells. It would be interesting to somewhat elaborate on the wider picture of P2Y10 expression T cells based on the findings of the authors and/or available literature.
3. The KO model is CD4-specific. Would it be possible to provide some comparisons with CD8+ T cells to further prove specificity of the actions of P2Y10? Fig 3D, Fig 5B-D, and Fig 7B would be particularly interesting in this respect as you would expect to see no difference here between WT and KO mice.

Minor points

- In panels 1K and 2E, hyphens are missing (IL-17, GM-CSF)
- Also in 2E, check the label FoxP3
- Page 6, line 6 and 23: I suggest replacing 'upregulated' by 'expressed' as I'm not sure whether the higher transcription levels relate to cellular maturation or to activation as part of the EAE model
- Page 7, line 4: IL-17
- Page 9, line 21: detected

Reviewer #2 (Remarks to the Author):

With a few exceptions the data presented are reasonably solid and tell a convincing and interesting story that: Chemokine signalling through Gi GCPR induces LysoPS and ATP release which signals through P2Y10 and G12/13 to induce RhoA dependent chemotaxis – focused on trafficking to lymphoid organs via homeostatic chemokines such as CCL19.

Major

Can authors provide some rationale for using homeostatic chemokines known for recruitment to LN and BM for the migration studies using both naïve and activated T-cells – rather than inflammatory chemokines that would be found within the CNS or skin. This is particularly important given the story they portray is about altered migration to peripheral tissues rather than lymphoid organs. Additional evidence showing that key findings hold true in response to inflammatory chemokines such as CXCL9, CXCL10 as well as CCL4 and e.g. CCL5 would further strengthen the story that P2Y10 is involved in directing migration during inflammatory conditions.

Figure 3 – data on directionality and velocity are missing – for those cells still able to undergo chemotaxis did they move in the direction of the chemokine; what velocity are they moving at? For panel E what is the p value for the basal migration – activation is known to increase the stickiness of the cells and reduce migration is this the case here? Please include the data akin to B-D for activated CD4 cells – are the same trends seen? Data for B-D, H, K-L are plotted as the mean of the cells per individual mouse; however, the statistical analysis has been done on the individual cells for all the mice and not the data presented within the graphs which has a high

degree of variability between groups with some mice demonstrating the same distance travelled or % immobile cells in both groups – are the data in the graphs, i.e. the mean of each mouse statistically different?

Polarisation studies are not convincing and based solely on phalloidin staining rather than true markers of polarisation and directionality of migration in these cells. Higher resolution images and additional markers are required such as RhoA and ROCK as indicated by the authors in their discussion.

Supp Fig 1 – Jurkats are not representative of circulating human peripheral blood lymphocytes with inherently different adhesive and migratory properties. Strongly suggest that these experiments (including Figure 3I) are replaced with some using healthy donors and even MS patient blood to truly understand the significance of this pathway in humans – these would be representative of the cells migrating into the CNS in MS. Additionally data in sFig 1 are missing protein expression with only 50% reduction in gene expression detected at the time point analysed. Linked to Figure 3I what is the level of migration in resting non-transfected Jurkats – the level of chemotaxis here is very low (>6%) compared to 30-40% when human PBL/T-cells are used.

Does blocking chemokine induced ATP/LysoPS abrogate human MS patient CD4 T-cell migration? i.e. is there any evidence the murine system translates to the patients where the therapy would be deployed?

Minor

Figure legend 1 - (J, n=7)) typo with two brackets; also K defines y axis as cytokine expressing cells but FOXP3 isn't a cytokine but rather a transcription factor.

Text indicates that RT-PCR was used in Figure 1D but the legend states sequencing and data are expressed as RPKM. Can authors clarify the methodology used for gene expression studies. Also methods indicates $\Delta\Delta C_t$ method was used but none of the figures report this?

Figure 2G – CD11b is not specific to macrophages; it is a major neutrophil marker – how are these cells gated.

Supp Fig 1 – x axis is not labelled; figure legend has no n value; also data are missing for the nucleofected and negative siRNA controls.

Figure 2 B and H – representative images would be appreciated; E – axis label reads IFoxP3 typo should be FoxP3

Figure 3 – B-D have no n values for the number of mice. In J please include an arrow highlighting the direction of the chemotactic gradient. Also confirm what % of input means in the figure legend as the input was PBMC and not naive CD4 cells.

Figure 5 – G the empty white bars on all the graphs are supposed to represent the control naive T-cells; but G is COS-1 cells – the colour should be changed to avoid confusion. Data for activated T-cells are missing; Figure 6F has the CCL19 induced ATP release, but no LysoPS data has been included.

Suppl Fig 2 – n are missing from legend; concentration of CCL19 1ug/ml doesn't match the concentration used in the main paper (100ng/ml).

Suppl Fig 3 – Labelling in the figure legend is wrong – “Expression of P2X and P2Y receptors in immature T cells (A) and granulocytes (B) from murine bone marrow (adapted from Tabula muris1, shown is mean expression of individual cells). ” Should be B and C. n are missing from legend

Figure 6 – E appears irrelevant to figure title and linked with Suppl Fig 3? Surely the expression of the P2X and P2Y receptors should be discussed earlier in the paper. Also the statistical analysis in

this figure contrasts the rest of the paper – where genotype expressions for even in vitro analyses have been performed using unpaired t-tests – but now these data are analysed using paired t-test. Can authors please explain the experimental design for the earlier in vitro studies.

Figure 7 – this is a mismatch of data and overview – A and B could be included into Figure 6, C and D could be included within Figure 2. An explanation is required for E in the figure legend.

Please include the data on memory T-cells from p6 lines 15-17 in the supplementary figures.

P10 and 43 title implies pannexin 1 involvement but CBX is a pan inhibitor of pannexin and connexins. The use of a specific pannexin 1 inhibitor is required to justify these titles; in the absence of this the titles need to be changed. The supplementary data shows some gene expression of other family members; and the absence of any protein expression data means the conclusions on p13 line 21-22 are limited. Gene and protein expression do not directly correlate – so it is important to show that CD4 T-cells only express pannexin 1 on their surface to make these statements.

Please expand the methods in the chemotaxis chamber to include how data in Fig 3 B-D were analysed.

Typo grammatical errors

P4 line 23 – ligands was should be ligands were

P5 line 2 comma after DCs should be deleted.

P6 line 13 should read “were not affected”

P7 line 7 FACS analysis implies the cells were sorted is this the case? If not this should read flow cytometry.

P9 line 5 – (IP1)) second bracket needs deleting

P9 line 7 – sentence linked to Fig 5E phrasing needs improving

P9 line 11 – again this sentence is repetitive – suggest deleting “in these cell already” and moving “in empty vector-transfected cells” to after “mobilisation”

P9 line 21 deteced should read detected

P10 line 7 – the word abolished seems too strong given there is still RhoA activation – the levels are not equivalent to baseline based the data presented.

P28 line 25 indicates paired test was used – however the majority of the stats in the figure legends is reported as unpaired test which is not mentioned in this section. Also please include a sentence on how normality was determined.

Reviewer #3 (Remarks to the Author):

In the manuscript entitled “G-protein-coupled receptor P2Y10 facilitates chemokine-induced CD4 T cell migration through an autocrine feedback loop involving ATP and lysophosphatidylserine” by Gurusamy et al., describes the generation and immunological characterization of P2Y10-null mice, and have tested these mice in an EAE model for assessing the effect of P2Y10 on neuroinflammation. Further the authors show by in vitro assays, that CD4 T cells deficient in P2Y10 have reduced migration potential and chemokine-induced RhoA activation. The authors also attempt to show that lysoPS and ATP enhance RhoA activity in P2Y10 dependent manner, and that ATP degradation reduces this effect. Lastly, the authors try to show that CD4 T cell migration in P2Y10-null mice is compromised in a hypersensitivity model. While the studies are interesting, the manuscript in the current form is quite preliminary, and has several major drawbacks, in view of this reviewer:

Effects from lysoPS & specificity of P2Y10 to lysoPS:

1. The authors start with the notion that P2Y10 is a lysoPS receptor, even though there is enough literature speculation on this matter, as mentioned by the authors in this manuscript. While the authors show that lysoPS induced RhoA activation in WT CD4 T cells, the effect seen in P2Y10-null cells, is at best modest. Therefore, it is absolutely necessary to assess this more carefully. It would

be useful to clearly test out a panel of all lysophospholipids other than S1P and lysoPA (e.g. lysoPI, lysoPT, lysoPG, lysoPE) to figure out the lysophospholipid specificity of P2Y10, and the effects of these may have on RhoA activation.

2. On a similar note, it is imperative to assess the concentrations of all these lysophospholipids in CD4T cells (lysates and supernatant) following CCL19 stimulation, and in tissues (at least serum, brain and spinal cord) of P2Y10-null mice, to ensure that effects are lysoPS specific, if at all.

3. Additionally, if these effects are indeed lysoPS specific, a recent report (in Cell Chemical Biology) shows that the lipid tail of lysoPS affects different immunological phenotypes. In light of this, it would be useful for the authors to actually measure lysoPSs of different lipid tails in cultured CD4 T cells +/- stimulation to ascertain which of these might be producing the effects (if any).

EAE model studies:

1. Based on the neurological scoring chart (figures 2A, 2C), it seems that the early disease progression (days 10 – 12) is more for the P2Y10-null mice. Therefore, one could argue that equal number of CD4 T cells for both genotypes are infiltrating the CNS, and perhaps P2Y10-null CD4 T cells are unable to survive long enough to produce the pathological effects in the CNS. How do the authors rule out this possibility? The author should do early time points measurements day 5, day 10, day 15 to rule this out.

2. Does the tone of serum or CNS lysoPS or other lysophospholipids change during the course of disease progression? Does this correlate with CD4 T cells infiltration in the CNS?

3. A similar effect is seen with GPR174-null mice. What is the functional redundancies between these GPCRs? What is the cross-talk between them in this model?

Autocrine model:

1. The authors propose an autocrine model for stimulation of CD4 T cells by lysoPS and ATP. This model in the eyes of the reviewer is far-fetched, and based purely on the in vitro assays. It is known that in response to stimuli, macrophages and mast cells too secrete significant lysoPS, and may be the specific effects from lysoPS (if proved) may in fact be paracrine?

2. The authors can further try supplying non-hydrolysable forms of ATP to discern whether endogenous ATP is needed for activating P2Y10, and the ATP aspect of autocrine signaling may also be worked out from these studies?

3. The authors can also try to co-culture LPS stimulated macrophages with naïve CD4 T cells, and see if macrophages stimulation causes stimulation of CD4 T cells for instance?

LysoPS and neuroinflammation:

1. LysoPSs are known to induce neuroinflammation, especially very-long chain components, in the context of the neurological disorder PHARC. Yet, while the authors discuss about lysoPS and neuroinflammation, it is disappointing that the authors don't discuss this pertinent aspect at all. On this note, it would be good to know whether ABHD12 has any effect on lysoPS in CD4 T cells, and if P2Y10-depletion can rescue neuroinflammation in ABHD12-null mice. ABHD12 inhibitors are described in literature, and the authors can perhaps use these to address this concern.

2. Also, lysoPS/ABHD12 has shown to be involved in T-cell mediated LCMV challenges, and it would be good for the authors to discuss this point in the context of their studies as well, and prospect on the role of P2Y10 (and GPR34) in regulating these.

EDITOR REMARKS

In particular, you need to establish the lysophospholipid specificity of P2Y10 in the context of RhoA, and the concentration of the candidate ligands in wild type and P2Y10 Cd4 T cells upon Ccl9 stimulation, as per reviewer #3. It is important that you demonstrate the effect of inflammatory chemokines, that are more pertinent to the disease model than homeostatic chemokines, in the migration experiments. We require that you expand your study to human samples, as reviewer #2 suggests. Paracrine regulatory mechanisms in the EAE model should be considered, as per reviewer #3. Wider context regarding the role of LysoPSs in neuroinflammation must be provided. Lastly, please comprehensively address the referees' other concerns.

REVIEWER COMMENTS

Reviewer #1 (Remarks to the Author):

Provided is a careful investigation of P2Y10 in CD4+ T cells describing its expression, role in cell migration, (autocrine) mechanism of action, and effect in the EAE model of neuroinflammation. Experiments are novel, informative, well conducted, and clearly described.

I have a few comments:

1. The link with MS provided in the Introduction reads somewhat exaggerated and is not necessary for understanding the context or appreciating the value of this study. I suggest shortening this part or providing data that indeed relate to MS.

Following the reviewer's advice we have removed all information that is not truly relevant in the context of this study, for example regarding prevalence and therapeutic aspects of MS.

2. Fig 1A compares GPCR expression between naïve and spinal cord-infiltrating CD4+ T cells. Has GPCR expression also been measured in mature, circulating CD4+ T cells? And how about CD8+ T cells? In our own, unpublished data of human brain-resident and paired circulating T cells, we notice that the P2RY10 gene is expressed more abundantly than other P2X/P2Y receptor gene and at comparable levels in CD4+ vs CD8+ as well as brain- vs blood-derived T cells. It would be interesting to somewhat elaborate on the wider picture of P2Y10 expression T cells based on the findings of the authors and/or available literature.

This is an interesting question, unfortunately our initial Nanostring-based expression analysis did not include peripheral blood T cells. To nevertheless address the reviewer's question, we analyzed published gene expression data sets. Analysis of single-cell expression data set GSE108097¹ showed that *P2ry10* is also in blood lymphocytes the most abundantly expressed purinergic receptor, both in CD4 T cells and in CD8 T cells (Figure 1).

Figure 1: Purinergic receptor expression in murine peripheral blood CD4 and CD8 T cells as determined by single-cell RNAseq (Data re-analyzed from GSE 108097¹; Analysis of 100 CD4-positive and 659 CD8-positive T cells from peripheral blood of 4 healthy mice).

To test whether P2Y10 expression in peripheral blood CD4 T cells changed during EAE, we isolated circulating CD4 and CD8 T cells from healthy mice and mice at peak EAE disease. We found that both in CD4 and CD8 T cells *P2ry10* expression was significantly increased (Figure 2).

Figure 2: *P2ry10* expression in CD4 and CD8 T cells isolated by magnetic cell separation from peripheral blood of healthy mice (basal) or mice at peak disease of EAE (d20) (n=2-4 mice each, data are normalized to *Gapdh*). *, $P < 0.05$.

We also analyzed expression of purinergic receptors in human blood CD4 cells and found that *P2RY10* ranked third with respect to overall expression strength, only surpassed by *P2RY8* (which does not have a mouse orthologue) and *P2RX4* (Figure 3, data adapted from GSE78244)².

Figure 3: Expression of P2X and P2Y receptors in human blood CD4 T cells (adapted from GSE78244 ², shown is mean gene expression).

We show the data on purinergic receptor expression in murine blood CD4 T cells in Suppl. Figure 6B of the revised manuscript and describe them on page 10, lines 9-11. In addition, we show the data on the upregulation of P2Y10 in EAE in Suppl. Figure 1 and describe them on page 6, lines 7-8. A statement regarding the potential role of P2Y10 in CD8 T cells was included in the discussion on page 14, lines 1-2 (see also our response to point 3).

3. *The KO model is CD4-specific. Would it be possible to provide some comparisons with CD8+ T cells to further prove specificity of the actions of P2Y10? Fig 3D, Fig 5B-D, and Fig 7B would be particularly interesting in this respect as you would expect to see no difference here between WT and KO mice.*

This is an interesting idea, unfortunately recombination in the CD4-Cre line is not completely restricted to mature CD4 cells due to the fact that all T cells have to pass the CD4/CD8 double positive state in the thymus. This short period of developmental CD4 expression suffices to cause a certain degree of recombination in those cells that will later develop into CD8 cells ³. Accordingly, we observed some reduction of P2Y10 immunoreactivity in CD8 cells, while there is no reduction in B cells (Figure 4).

Figure 4: Reduction of P2Y10 expression in different lymphocyte populations isolated from CD4-P2Y10-KO mice as judged by immunoblotting (A, exemplary immunoblots, B, statistical evaluation). GAPDH as loading control (n=2-5). ***, $P < 0.001$.

We tested whether reduced P2Y10 expression would impair migration in CD8 cells obtained from CD4-P2Y10-KOs, and found that this was indeed the case, though the level of reduction was lower than in CD4 T cells (Figure 5A). However, since the reviewer’s question aimed at a testing migration in a lymphocyte population not affected by Cre-mediated deletion of P2Y10, we tested chemokine-induced migration in B cells. As shown in Figure 5B, B cells isolated from CD4-P2Y10-KOs showed normal migration SDF-1 α and CCL19.

Figure 5: Transwell migration of naïve CD4 and CD8 T cells (A) and of B cells (B) under basal conditions and after addition of 100 ng/ml of the indicated chemokines to the lower well (n=4-6). *, $P < 0.05$.

These data show that due to transient expression of CD4Cre in double positive thymocytes, also CD8 cells show reduced levels of P2Y10 expression in CD4-P2Y10-KOs. As in CD4 T cells, loss of P2Y10 in CD8 cells results in impaired chemokine-induced migration, though the impairment is less prominent than in CD4 T cells. In line with this, spinal cord infiltration with CD8 cells is not significantly reduced in the EAE model (see Figure 2D of the revised manuscript). We included the following statement regarding the potential role of P2Y10 in CD8 T cells on page 14, lines 1-6:

“Due to transient Cre expression in the CD4/CD8 double positive stage, also mature CD8 cells show a trend to reduced P2Y10 expression, and chemokine-induced migration was tendentially reduced in these cells (data not shown). However, this effect was not strong enough to cause a significant reduction of CD8 T cell infiltration into the spinal cord during EAE (see Fig. 2D).”

Minor points

- In panels 1K and 2E, hyphens are missing (IL-17, GM-CSF)

The missing hyphens have been added.

- Also in 2E, check the label FoxP3

The superfluous "l" has been removed.

- Page 6, line 6 and 23: I suggest replacing 'upregulated' by 'expressed' as I'm not sure whether the higher transcription levels relate to cellular maturation or to activation as part of the EAE model

'upregulated' has been replaced by 'expressed'.

- Page 7, line 4: IL-17

The missing hyphen has been added.

- Page 9, line 21: detected

The typing error has been corrected.

Reviewer #2 (Remarks to the Author):

With a few exceptions the data presented are reasonably solid and tell a convincing and interesting story that: Chemokine signalling through Gi GPCR induces LysoPS and ATP release which signals through P2Y10 and G12/13 to induce RhoA dependent chemotaxis – focused on trafficking to lymphoid organs via homeostatic chemokines such as CCL19.

Major

1. Can authors provide some rationale for using homeostatic chemokines known for recruitment to LN and BM for the migration studies using both naïve and activated T-cells – rather than inflammatory chemokines that would be found within the CNS or skin. This is particularly important given the story they portray is about altered migration to peripheral tissues rather than lymphoid organs. Additional evidence showing that key findings hold true in response to inflammatory chemokines such as CXCL9, CXCL10 as well as CCL4 and e.g. CCL5 would further strengthen the story that P2Y10 is involved in directing migration during inflammatory conditions.

Following the reviewer’s suggestion we performed additional migration assays, this time employing inflammatory chemokines CXCL9, CXCL10, CCL4 and CCL5. As observed for the homeostatic chemokines, migration was also in response to these chemokines reduced in P2Y10-deficient CD4 T cells compared to control CD4 T cells (Figure 6).

Figure 6: Transwell migration of naïve CD4 T cells (nCD4) under basal conditions and after addition of 100 ng/ml of the indicated chemokines to the lower well (data from 8 mice per group). *, $P<0.05$; **, $P<0.001$; ***, $P<0.001$.

We added these data to Figure 3A of the revised manuscript and report them on page 7, lines 17-18.

2. Figure 3 – data on directionality and velocity are missing – for those cells still able to undergo chemotaxis did they move in the direction of the chemokine; what velocity are they moving at?

As suggested by the reviewer, we determined directionality and velocity during CCL19-induced migration in μ -slide chemotaxis chambers. We found that velocity and distances travelled were reduced in P2Y10-deficient CD4 T cells, but not the directness of migration (Figure 7).

Figure 7: Live cell imaging analysis of migratory behavior in CCL19-stimulated CD4 T cells: Exemplary tracking result (A) and statistical evaluation of accumulated distance (B), velocity (C), and directness (D) (3-5 mice per group were analyzed in 4-8 independent tracking experiments, and per mouse 937-7969 cells were evaluated). Data are means \pm SEM; comparisons between genotypes were performed using unpaired Student's *t*-test. *, $P < 0.05$.

These data have been included in Figures 3C-F of the revised manuscript and are described on page 7, lines 19-21.

For panel E what is the p value for the basal migration – activation is known to increase the stickiness of the cells and reduce migration is this the case here?

Indeed, the difference in basal migration is significant (Figure 8), and the missing p value has been added to former Figure 3G (now 3I).

Figure 8: Transwell migration of CD4 cells that have been unspecifically activated by CD3/CD28 stimulation (aCD4) (E, $n = 7-8$). *, $P < 0.05$.

To test whether the basal difference in aCD4 migration is secondary to altered adhesion to cell culture plastic, we determined the proportion of activated CD4 cells firmly attached after 30 minutes of culture. In both genotypes approximately 15% of cells were firmly attached, suggesting there was no major difference in general adhesiveness in P2Y10-deficient cells (Figure 9).

Figure 9: Adhesion of control and P2Y10-deficient CD4 T cells to cell culture plastic. Isolated CD4 T cells were stimulated for 3 days with CD3/CD28 beads, then 1×10^5 cells per well were allowed to adhere in 96 well plates for 30 min at $37^\circ\text{C} / 5\% \text{CO}_2$. Nonadherent cells were removed by two rounds of gentle washing with PBS, and adherent cells were detached in RPMI 1640 / 5 mM EDTA for 20 min on ice. Cells were stained for CD4 and the absolute number of adherent CD4 was determined in relation to the absolute number of CD4 cells in the input sample.

We conclude that the observed difference in basal migration of activated CD4 T cells is not due to altered adhesiveness. Instead, we hypothesize that the basal difference in migration is the consequence of an autocrine release of chemokines from activated CD4 T cells, which then facilitates P2Y10-dependent RhoA activation, polarization, and migration even in the absence of exogenous chemokine stimulation. In line with this hypothesis, we found that activated CD4 T cells of both control and KO mice release considerable amounts of chemokines (Figure 10).

Figure 10: Concentrations of chemokines in the supernatant of unspecifically activated CD4 T cells from control and KO mice were determined by multiplex bead array assay (5-6 mice per group).

We added a statement regarding difference in basal migration in activated CD4 cells on page 9, lines 5-7.

Please include the data akin to B-D for activated CD4 cells – are the same trends seen?

In analogy to former Figure 3D (which is now Figure 3B, the old Figures 3B and 3C have been replaced in the revised version of the manuscript), we determined the proportion of immobile cells in unspecifically activated CD4 T cells from both genotypes and found also here an increased percentage of immobile cells (Figure 11).

Figure 11: Live cell imaging analysis of migratory behavior of activated CD4 T cells in response to CCL19 stimulation: Quantification of immobile versus mobile cells (cells from 3 mice per group, per experiment 44-83 cells were evaluated). *, $P < 0.05$.

We show these data in Suppl. Figure 3A of the revised manuscript and mention them on page 7, line 24.

Data for B-D, H, K-L are plotted as the mean of the cells per individual mouse; however, the statistical analysis has been done on the individual cells for all the mice and not the data presented within the graphs which has a high degree of variability between groups with some mice demonstrating the same distance travelled or % immobile cells in both groups – are the data in the graphs, i.e. the mean of each mouse statistically different?

We acknowledge that the description of our analyses in Figure 3 is confusing and would therefore like to clarify as follows:

- In former Figures 3D, K, L (now Figures 3B, 4B, 4C), each data point represents the percentage of CD4 T cells displaying a certain behavior (immobile or polarized, respectively) in one independent experiment.
 - o In former Figure 3D/new Figure 3B, cells from 3 mice per group were analyzed in 5 independent experiments, and per experiment 64-200 cells were evaluated.

- In former Figures 3K-L/new Figures 4B-C, cell from 4-5 mice per group were analyzed in 4-5 independent experiments, and per experiment 57-259 cells (nCD4) and 32-230 cells (aCD4) were evaluated.
- In former Figure 3H (now Figure 3L), the data points represent the percentage of CD4 T cells displaying a certain behavior (P2Y10 expression) in a previously published single cell expression analysis.
- In former Figures 3B-C, we showed one example of three independent migration experiments, but these data have been replaced by a new analysis that includes accumulated distance, velocity, and directness (new Figures 3C-F). In this new data set, cells from 3-5 mice per group were analyzed in 4-8 independent tracking experiments, and per experiment 937-7969 cells were evaluated.

Information regarding total number of mice, independent experiments, and individual cells per experiment has been added to the Figure legend of revised Figures 3 and 4.

3. Polarisation studies are not convincing and based solely on phalloidin staining rather than true markers of polarisation and directionality of migration in these cells. Higher resolution images and additional markers are required such as RhoA and ROCK as indicated by the authors in their discussion.

Following the reviewer's suggestion we included as an additional marker for T cell polarization the activation of small GTPase RhoA, which has been shown to be activated at the leading and trailing edge of migrating T cells⁴. To label active RhoA, we employed the GST-tagged RhoA binding domain of rhotekin, which binds specifically to GTP-bound, active RhoA and can be detected by anti-GST immunofluorescence staining^{5,6}. Of note, since GST-RBD staining of active RhoA did not work well in μ -slide chemotaxis chambers, these analyses were done on normal cell culture plastic in a homogeneous CCL19 field. These analyses showed that the proportion of cells with an enrichment of active RhoA at the leading and/or trailing edge was significantly reduced in P2Y10-deficient CD4 T cells (Figure 12).

Figure 12: Distribution of active RhoA in CCL19 (10 min)-stimulated CD4 T cells: **A**, Exemplary photomicrographs showing distribution of active RhoA and phalloidin staining in CCL19-polarized control cells. **B,C**, Exemplary photomicrographs (B) and

statistical evaluation (C) of cells with polarized RhoA activation (n=492 and 302 cells from 2 independent experiments; * indicates cells with polarized RhoA activation).
 ***, $P < 0.001$.

We added these data to Figure 4D-F of the revised manuscript and report them on page 8, lines 17-24.

4. *Supp Fig 1 – Jurkats are not representative of circulating human peripheral blood lymphocytes with inherently different adhesive and migratory properties. Strongly suggest that these experiments (including Figure 3I) are replaced with some using healthy donors and even MS patient blood to truly understand the significance of this pathway in humans – these would be representative of the cells migrating into the CNS in MS.*

We agree that Jurkat cells are a poor substitute for primary human CD4 T cells. Following the reviewer’s advice, we established siRNA-mediated knockdown in primary CD4 T cells obtained from healthy human donors and MS patients. Treatment with P2Y10-specific siRNAs (Accell siRNA, Horizon Discovery) resulted compared to treatment with scramble siRNA in a reduction of *P2RY10* transcript levels to approximately 40% (Figure 13A). On the protein level the reduction was comparable, though with higher variation (Figure 13B). This reduction of P2Y10 levels was sufficient to cause significant impairment of SDF-1 α -induced migration in CD4 T cells isolated from peripheral blood of healthy donors (Figure 13C, left). To test whether P2Y10 knockdown had comparable effect in MS patients, we isolated CD4 T cells from patients that had been diagnosed with MS and were currently without treatment (but had received different types of immunotherapy in the past: 1 patient > 3 months after ocrelizumab, 1 patient > 6 months after ocrelizumab, 1 patient > 3 months after corticosteroid pulse therapy). As observed in CD4 T cells from healthy donors, knockdown of P2Y10 resulted in reduction of SDF-1 α -induced migration (Figure 13C, right).

Figure 13: Chemokine-induced migration in primary human CD4 T cells after P2Y10 knockdown. **A, B,** P2Y10 expression in human CD4 T cells obtained from peripheral blood was determined after treatment with scramble siRNA (siContr) or siRNA directed against P2Y10 (siP2Y10) by RT-PCR (A, n=4) and western blotting (B, n=2). **C,** Transwell migration of naïve human CD4 T cells isolated from healthy donors (left) or

MS patients (right) under basal conditions and after addition of 100 ng/ml SDF-1 α to the lower well (n=3). *, $P < 0.05$; **, $P < 0.01$.

We show these data in Figure 3M and Suppl. Figure 3B of the revised manuscript and describe them on page 8, lines 4-12.

Additionally data in sFig 1 are missing protein expression with only 50% reduction in gene expression detected at the time point analysed. Linked to Figure 3I what is the level of migration in resting non-transfected Jurkats – the level of chemotaxis here is very low (>6%) compared to 30-40% when human PBL/T-cells are used.

We replaced Jurkat data by the abovementioned data obtained in primary CD4 T cells from MS patients (see above).

Does blocking chemokine induced ATP/LysoPS abrogate human MS patient CD4 T-cell migration? i.e. is there any evidence the murine system translates to the patients where the therapy would be deployed?

Following the reviewer's suggestion we determined the effect of apyrase on chemokine-induced migration in primary CD4 T cells from healthy donors and MS patients (1 patient without treatment, 1 patient > 3 months after corticosteroid pulse therapy, 1 patient receiving interferon beta-1b treatment). We found that both groups showed clear reduction of SDF-1 α -induced migration in the presence of apyrase (Figure 14).

Figure 14: Effect of apyrase pretreatment on SDF-1 α -induced transwell migration in naïve human CD4 T cells isolated from peripheral blood of healthy donors or MS patients (n=3). *, $P < 0.05$.

We show these data in Figure 6C of the revised manuscript and discuss them on page 11, lines 14-15.

Minor

Figure legend 1 - (J, n=7)) typo with two brackets; also K defines y axis as cytokine expressing cells but FOXP3 isn't a cytokine but rather a transcription factor.

The superfluous bracket was removed. In K we replaced the y axis label “Cytokine expressing ...” by “Cytokine / FoxP3 expressing ...”; in the Figure legend we replaced “cytokine-expressing cells” by “IFN γ / IL-17A / FoxP3-expressing cells”.

Text indicates that RT-PCR was used in Figure 1D but the legend states sequencing and data are expressed as RPKM. Can authors clarify the methodology used for gene expression studies. Also methods indicates $\Delta\Delta Ct$ method was used but none of the figures report this?

The data shown in Figure 1D are indeed from mRNA sequencing, not from RT-PCR, and we changed the text accordingly. The initial labeling as RPKM was incorrect and has been replaced by the term “library-sized normalized counts” throughout the manuscript (also Figures 3G,H of revised manuscript as well as Suppl. Figures 6A,F).

Figure 2G – CD11b is not specific to macrophages; it is a major neutrophil marker – how are these cells gated.

The population designated as “CD11b-pos; CD45high” was gated as follows: FSC/SSC \rightarrow gate on living single cells \rightarrow CD45pos \rightarrow CD11b-pos/Ly6g-neg \rightarrow CD45high. Figure 15 shows an example of the gating strategy. To clarify this issue in the text, we added the term “Ly6G^{neg}” to the description of those cells both in the revised text (page 7, line 9) and the revised Figure 2G (now “CD11b^{pos}, Ly6G^{neg}, CD45^{hi} macrophages”).

Figure 15: Gating strategy for the differentiation of myeloid subtypes in spinal cord of EAE mice.

Supp Fig 1 – x axis is not labelled; figure legend has no n value; also data are missing for the nucleofected and negative siRNA controls.

The x axis labeling and n number for former Suppl. Figure 1 (now Suppl. Figure 4) has been added. The figure shows now knockdown of P2Y10 in primary human CD4 T cells instead of Jurkat cells, scramble siRNA was used for the control group.

Figure 2 B and H – representative images would be appreciated;

As suggested by the reviewer, we included examples of the luxol fast blue staining and Mac3 staining in the Figures 2B and 2H of the revised manuscript (see also Figure 16 below).

Figure 16: A, Quantification of spinal cord demyelination by luxol fast blue (LFB) staining on day 28 (B) after induction of EAE: Exemplary photomicrograph (left) and statistical evaluation (right) (n=8). **B,** Quantification of Mac3-positive cells in lumbar CNS (H): Exemplary photomicrograph (left) and statistical evaluation (right) (n=8). Data are means \pm SEM; comparisons between genotypes were performed using unpaired Student's *t*-test. *, $P < 0.05$; **, $P < 0.01$.

E – axis label reads *IFoxP3 typo should be FoxP3*

The typing error has been corrected.

Figure 3 – B-D have no n values for the number of mice.

In these experiments, cells from 3 mice per group were analyzed in 5 independent experiments. This information has been added to the legend of revised Figure 3B (former Figure 3D); the former Figures 3B/C have been replaced by new data.

In J please include an arrow highlighting the direction of the chemotactic gradient.

The direction of the chemotactic gradient is now indicated beneath the photomicrographs in revised Figure 4A as shown in Figure 17 below.

Figure 17: Phalloidin staining of CD4 T cells that were allowed to migrate for 10 minutes towards a CCL19 gradient in μ -Slide Chemotaxis chambers: representative photomicrographs.

Also confirm what % of input means in the figure legend as the input was PBMC and not naïve CD4 cells.

The term “% input” is in all cases calculated as follows:

$$\% \text{ input} = \frac{\text{Number of transmigrated CD4 T cells}}{\text{Number of CD4 T cells added to the upper well}} \times 100$$

In most cases, transmigration assays were performed with MACS-isolated CD4 T cells, only the effect of apyrase on human CD4 T cell migration (Figure 6C) was investigated in total PBMCs. A statement regarding the calculation of % input as well as of the use of isolated CD4 vs total PBMC was added to the methods section on page 24, line 9 and 18-19.

Figure 5 – G the empty white bars on all the graphs are supposed to represent the control naïve T-cells; but G is COS-1 cells – the colour should be changed to avoid confusion.

We have changed to colour-coding for Figure 5G.

Data for activated T-cells are missing; Figure 6F has the CCL19 induced ATP release, but no LysoPS data has been included.

As suggested by the reviewer, we determined levels of active RhoA in unspecifically activated CD4 T cells and found that they showed already under basal conditions reduced RhoA activation (Figure 18A). We furthermore found in a side-by-side comparison of naïve vs

activated CD4 T cells that levels of RhoA-GTP increase in response to activation in control CD4 T cells, but not so much in P2Y10-deficient cells (Figure 18B). This difference in basal RhoA activation might contribute to the difference in basal migration observed in P2Y10-deficient CD4 T cells (see also our response to this reviewer’s question 2 on pages 7 and 8 of this letter). As described above, we hypothesize that these basal differences are the consequence of autocrine release of chemokines from activated CD4 T cells (Figure 18C), which might facilitate P2Y10-dependent RhoA activation, polarization and migration even in the absence of exogenous chemokine stimulation.

Figure 18. A, Basal levels of active GTP-bound RhoA in activated CD4 T cells from control and CD4-P2Y10-KOs (6 mice per group). **B,** Comparison of levels of active RhoA levels in naïve and activated CD4 T cells from the same donor (3 mice per group). **C,** Concentrations of chemokines in the supernatant of unspecifically activated by CD4 T cells from control and KO mice were determined by multiplex bead array assay (5-6 mice per group).

We added information on basal RhoA activation and basal migration in activated CD4 T cells on page 9, line 4-7.

Regarding chemokine-induced production of LysoPS in activated CD4 T cells, we unfortunately forgot to include such samples in the extensive mass spectrometric analysis of lysophospholipid production requested by reviewer 3 (and noticed our mistake only very recently). Since waiting time for the existing mass spectrometry data had been nearly 3 months, and since most other studies have been done in naïve cells, we hope the reviewer finds this revision acceptable even without these additional mass spectrometry data.

Suppl Fig 2 – n are missing from legend; concentration of CCL19 1ug/ml doesn't match the concentration used in the main paper (100ng/ml).

We here used a concentration exceeding the normal dosage to show that not even at maximal CCL19 stimulation (10 times higher than normal), P2Y10-dependent responses can be elicited. We added a corresponding sentence to the figure legend to explain this choice.

Suppl Fig 3 – Labelling in the figure legend is wrong – “Expression of P2X and P2Y receptors in immature T cells (A) and granulocytes (B) from murine bone marrow (adapted from Tabula muris1, shown is mean expression of individual cells).” Should be B and C. n are missing from legend

The faulty figure legend has been adapted (now Suppl. Figures 6D and E) and n numbers (60 and 761 cells, respectively) have been added.

Figure 6 – E appears irrelevant to figure title and linked with Suppl Fig 3? Surely the expression of the P2X and P2Y receptors should be discussed earlier in the paper.

Following the reviewer’s suggestion we have transferred the data on purinergic receptor expression in CD4 T cells (former Figure 6E) to Suppl. Figure 6A and mention it now already in the context of Figure 5 (page 10, lines 7-9).

Also the statistical analysis in this figure contrasts the rest of the paper – where genotype expressions for even in vitro analyses have been performed using unpaired t-tests – but now these data are analysed using paired t-test. Can authors please explain the experimental design for the earlier in vitro studies.

Comparisons of two genotypes (control vs KO) on one variable were throughout the manuscript performed using unpaired t test. Experiments in which the same sample was analyzed under different conditions (e.g., treatment with/without apyrase) were analyzed using paired t test. We added a corresponding clarification in the methods section on page 28, line 15-16.

Figure 7 – this is a mismatch of data and overview – A and B could be included into Figure 6, C and D could be included within Figure 2. An explanation is required for E in the figure legend.

Following the reviewer’s advice, we integrated the carbenoxolon data (former Figure 7A) and the data on G_{12/13} (former Figure 7B) in Figure 6 (now Figure 6G and H). However, we would prefer to keep Figures 7C-E as a separate figure (now Figures 7A-C). As explanation for E (now C), we added the following statement:

“**C**, Schematic overview of findings: Chemokine stimulation of CD4 T cells results in the release of LysoPS and ATP, which in turn trigger P2Y10-dependent RhoA activation and consecutive facilitation of polarization and migration.”

Please include the data on memory T-cells from p6 lines 15-17 in the supplementary figures.

Data on regulatory and memory T cells are shown in Figure 19 and have been added to Suppl. Figure 2.

Figure 19: The percentage of FoxP3-positive regulatory T cells (A, C) or CD44⁺CD62L⁻ effector memory, CD44⁺CD62L⁺ central memory, or CD44⁺CD62L⁺ naïve T cells (B, D) was determined in CD4 T cells from inguinal lymph nodes (A,B) or spleen (C,D) (n=6-10). Data are means ± SEM; comparisons between genotypes were performed using unpaired Student's t test. n, number of individual mice.

P10 and 43 title implies pannexin 1 involvement but CBX is a pan inhibitor of pannexin and connexins. The use of a specific pannexin 1 inhibitor is required to justify these titles; in the absence of this the titles need to be changed. The supplementary data shows some gene expression of other family members; and the absence of any protein expression data means the conclusions on p13 line 21-22 are limited. Gene and protein expression do not directly correlate – so it is important to show that CD4 T-cells only express pannexin 1 on their surface to make these statements.

As suggested earlier by the reviewer, the carbenoxolon data have been integrated in Figure 6 and the title of Figure 7 is now reduced to “P2Y10-dependent facilitation of migration contributes to CD4-mediated responses outside the CNS”.

Regarding the expression of pannexins and connexins on the protein level, we reanalyzed raw data of a proteomics study in mouse CD4 T cells⁷ and found that in these cells only Pannexin 1 was detected, but not the other pannexins or any of the connexins (Figure 20).

Figure 20: Protein abundance of different pannexins and connexins in a proteomic analysis in murine CD4 T cells (reanalyzed from PXD012831⁷).

We included these data in Suppl. Figure 6G and mention them on page 14, line 26, to page 15, line 1 of the revised manuscript.

Please expand the methods in the chemotaxis chamber to include how data in Fig 3 B-D were analysed.

The original analysis of “distance travelled” was done using the tracking function in ImageJ, these data have now been replaced by a new analysis including velocity, directness and accumulated distance (Figures 3C-F of revised manuscript). A description of these analyses has been added to the methods section on page 25, lines 13-20.

For the determination of the proportion of immobile cells, individual cells were manually tracked during a 20 min time-lapse movie (7 frames / min) and classified as follows: 1) immobile, not polarized, 2) stationary but polarizing, 3) polarizing and migrating, 4) only passive movement (floating). The proportion of immobile, not polarized cells (category 1) was expressed as the percentage of all analyzed cells within the same view field. We added a corresponding description to the methods section on page 24, line 24, to page 25, line 3.

Typo grammatical errors

P4 line 23 – ligands was should be ligands were

Changed as suggested.

P5 line 2 comma after DCs should be deleted.

Changed as suggested.

P6 line 13 should read “were not affected”

Changed as suggested.

P7 line 7 FACS analysis implies the cells were sorted is this the case? If not this should read flow cytometry.

Changed as suggested.

P9 line 5 – (IP1)) second bracket needs deleting

Superfluous bracket was deleted.

P9 line 7 – sentence linked to Fig 5E phrasing needs improving

The misspelled “where” was corrected to “were”.

P9 line 11 – again this sentence is repetitive – suggest deleting “in these cell already” and moving “in empty vector-transfected cells” to after “mobilisation”

Changed as suggested.

P9 line 21 detecetd should read detected

The typo was corrected.

P10 line 7 – the word abolished seems too strong given there is still RhoA activation – the levels are not equivalent to baseline based the data presented.

Changed to “significantly reduced”.

P28 line 25 indicates paired test was used – however the majority of the stats in the figure legends is reported as unpaired test which is not mentioned in this section. Also please include a sentence on how normality was determined.

The term “unpaired” was added to the methods section on page 28, line 15, and information regarding normality testing (Kolmogorov-Smirnov) was added on page 28, in line 19.

Reviewer #3 (Remarks to the Author):

Effects from lysoPS & specificity of P2Y10 to lysoPS:

1. The authors start with the notion that P2Y10 is a lysoPS receptor, even though there is enough literature speculation on this matter, as mentioned by the authors in this manuscript. While the authors show that lysoPS induced RhoA activation in WT CD4 T cells, the effect seen in P2Y10-null cells, is at best modest. Therefore, it is absolutely necessary to assess this more carefully. It would be useful to clearly test out a panel of all lysophospholipids other than S1P and lysoPA (e.g. lysoPI, lysoPT, lysoPG, lysoPE) to figure out the lysophospholipid specificity of P2Y10, and the effects of these may have on RhoA activation.

Following the reviewer's suggestion, we determined RhoA activation in response to lysophosphatidylethanolamine (LPE), lysophosphatidylglycerol (LPG) and lysophosphatidylinositol (LPI) in CD4 T cells from control and CD4-KO mice. The effects of these lysophospholipids were mild and did not differ between control and KO CD4 T cells (Figure 21).

Figure 21: RhoA activation in response to LPE, LPG, or LPI (1 μ M, 1 min) in control CD4 T cells and P2Y10-deficient CD4 T cells (4 mice per group).

These data are in line with previous studies suggesting that LPG, LPE, and LPI are not P2Y10 ligands⁸. We show these data in Supplemental Figure 5B and describe and discuss them on page 9, lines 24 to 26.

2. On a similar note, it is imperative to assess the concentrations of all these lysophospholipids in CD4T cells (lysates and supernatant) following CCL19 stimulation, and in tissues (at least serum, brain and spinal cord) of P2Y10-null mice, to ensure that effects are lysoPS specific, if at all.

As suggested by the reviewer, we determined concentrations of other lysophospholipids in cell lysates and supernatant following CCL19 stimulation (Figure 22). LPI and LPG were not reliably detected, but several LPC and LPE species were present in pellets and supernatants. However, neither LPC nor LPE species showed significant changes after 1, 3, or 10 minutes of CCL19 stimulation in control or KO cell pellets or supernatants (Figure 22).

Figure 22: Lysophospholipid species were determined by LC-MS/MS in lysates (A, B) and supernatants (C, D) of 5×10^5 isolated murine CD4 T cells from control mice and CD4-P2Y10-KOs at 0, 1, 3 and 10 minutes of CCL19 (100 ng/ml) stimulation (n=3).

We mention these findings on page 10, line 26, to page 11, line 3.

We also determined concentrations of other lysophospholipids in serum, brain and spinal cord of healthy controls and P2Y10-null mice, but did not find differences between the genotypes (Figures 23-25):

Figure 23: Lysophospholipid species were determined by LC-MS/MS in spinal cords of control mice and CD4-P2Y10-KOs (n=4).

Figure 24: LysoPL species were determined by LC-MS/MS in brains of control mice and CD4-P2Y10-KOs (n=4).

Figure 25: LysoPL species were determined by LC-MS/MS in sera of control mice and CD4-P2Y10-KOs (n=4) (LysoPS, LPI, LPG were not consistently detected).

We show these data in Suppl. Figure 8 and mention them on page 11, lines 5-7.

3. Additionally, if these effects are indeed lysoPS specific, a recent report (in Cell Chemical Biology) shows that the lipid tail of lysoPS affects different immunological phenotypes. In light of this, it would be useful for the authors to actually measure lysoPSs of different lipid tails in cultured CD4 T cells +/- stimulation to ascertain which of these might be producing the effects (if any).

In addition to LysoPS species 18:0 and 18:1, we found LysoPS 16:0 to be upregulated in CCL19 stimulated CD4 T cells from both control and KO (Figure 26). However, since the concentration of 16:0 is relatively low in spinal cord (and not upregulated in EAE, see Figure 29), we did not follow up on the biological function of 16:0.

Figure 26: LysoPS 16:0 was detected by LC-MS/MS in in lysates of CCL19-treated CD4 T cells from control mice and CD4-P2Y10-KOs (n=3).

We have included a statement regarding the upregulation of LysoPS 16:0 on page 10, line 26.

EAE model studies:

4. Based on the neurological scoring chart (figures 2A, 2C), it seems that the early disease progression (days 10 – 12) is more for the P2Y10-null mice. Therefore, one could argue that equal number of CD4 T cells for both genotypes are infiltrating the CNS, and perhaps P2Y10-null CD4 T cells are unable to survive long enough to produce the pathological effects in the CNS. How do the authors rule out this possibility? The author should do early time points measurements day 5, day 10, day 15 to rule this out.

It is true that control mice and P2Y10 KOs show similar neurological scores in the early disease phase, and only differ in the intermediate and late phases of EAE. Following the reviewer's suggestion, we analyzed numbers and viability of spinal cord-infiltrating CD4 T cells in control and KO mice at d5, d10, and d15 after disease induction. At these early time points, numbers of spinal cord-infiltrating CD4 cells did not differ significantly (Figure 27A),

and staining with Annexin V and 7AAD did not reveal significant differences in the proportion of early apoptotic or late apoptotic/necrotic cells (Figure 27 B, C). These findings show that at onset of EAE, CD4 T cell infiltration into the spinal cord is indeed normal in CD4-P2Y10-KOs, and that the reduction of infiltrating cells observed at later time points cannot clearly be attributed to differences in CD4 T cell survival.

Figure 27: A, Numbers of spinal cord-infiltrating CD4 T cells at days 5, 10, and 15 after EAE induction in control mice and CD4-P2Y10-KOs (n=5-7). **B, C,** Proportion of early apoptotic (Annexin V-positive, 7-AAD-negative) and late apoptotic (Annexin V-positive, 7-AAD-positive) CD4 T cells at days 5, 10, and 15 after EAE induction (n=5-7).

We mention these findings on page 7, line 7-8 of the revised manuscript.

Instead, we hypothesize that the selective relevance of P2Y10 in later phases of EAE development is due to a differential role of P2Y10 in Th1 and Th17 cells. Th17 infiltration has been reported to proceed Th1 infiltration during EAE development⁹, and our data suggest that loss of P2Y10 results in significantly reduced migration in Th1-differentiated cells, whereas migration of Th17-differentiated cells is less affected (Figure 28A, B). This finding is most likely due to differences in P2Y10 expression levels in these cells (Figure 28C). We therefore hypothesize that early infiltration with Th17 cells is largely normal in P2Y10-KOs, whereas later infiltration of Th1 cells is impaired, resulting in partial protection in the intermediate and late phase of the disease.

Figure 28: Transwell migration of CD4 cells that have been differentiated towards Th1 (A, n=3-7) or Th17 (B, n=7). **C,** Percentage of *P2ry10*-expressing cells among Th1- and Th17-differentiated lymph node CD4 T cells (reanalysis of single-cell expression data from ¹⁰).

We show these data in Figures 3J-L of the revised manuscript and discuss them on page 17, lines 2-6.

5. Does the tone of serum or CNS lysoPS or other lysophospholipids change during the course of disease progression? Does this correlate with CD4 T cells infiltration in the CNS?

Following the reviewer's suggestion, we determined LysoPS and other lysophospholipids at days 0, 5, 10, and 15 of EAE development. We found that LysoPS species 18:0, 18:1 and 22:6 are significantly upregulated on day 10, whereas species 16:0, 20:4, and 22:5 did not change (Figure 29A).

Figure 29: LysoPS species were determined by LC-MS/MS in spinal cords of control mice at time points 0, 5, 10, 15 after EAE induction (4 mice per group). Data are means \pm SEM; comparisons between time points were performed using two-way ANOVA with Dunnett's multiple comparisons test. **, $P < 0.01$; ***, $P < 0.001$.

Also other lysophospholipids showed increased expression on d10 after EAE induction, most prominently among them LPI 18:0, LPG 18:1, LPC 16:0 and 18:0, and LPE 18:0 and 18:1 (Figure 30). This upregulation coincides with the peak of T cell infiltration, but which of these changes are causal for immune cell infiltration and which are rather bystander effects, is unclear.

Figure 30: Lysophospholipid species were determined by LC-MS/MS in spinal cords of control mice at time points 0, 5, 10, 15 after EAE induction (4 mice per group). Data are means \pm SEM; comparisons between time points were performed using two-way ANOVA with Dunnett's multiple comparisons test. *, $P < 0.05$; **, $P < 0.01$; ***, $P < 0.001$.

We show the data on lysophospholipid levels in spinal cord during EAE in Suppl. Figure 7 and mention them on page 11, lines 3-5 of the revised manuscript.

In the serum, only LysoPC and LysoPE species were reliably detected, and they did not change significantly in response to EAE induction (Figure 31).

Figure 31: Lysophospholipid species were determined by LC-MS/MS in serum of control mice at time points 0, 5, 10, 15 after EAE induction (4 mice per group).

6. A similar effect is seen with GPR174-null mice. What is the functional redundancies between these GPCRs? What is the cross-talk between them in this model?

Indeed, global Gpr174 knockout mice also show reduced EAE. Barnes et al.¹¹ showed that LysoPS can act *in vitro* via GPR174 to suppress T cell proliferation and regulatory T cell generation, resulting in reduced EAE severity. We investigated whether similar effects were observed in P2Y10-deficient CD4 T cells, but did not find evidence for reduced LysoPS-mediated suppression of proliferation of CD4 T cells. One potential explanation for the

diverse biological functions of the two receptors is their differential coupling, since P2Y10 has been suggested to signal through heterotrimeric G-proteins of the G_{12/13} family¹², whereas Gpr174 effects require G_s¹³. We discuss these differences between GPR174 and P2Y10 on page 15, line 24, to page 16, line 4.

Autocrine model:

7 The authors propose an autocrine model for stimulation of CD4 T cells by lysoPS and ATP. This model in the eyes of the reviewer is far-fetched, and based purely on the in vitro assays. It is known that in response to stimuli, macrophages and mast cells too secrete significant lysoPS, and may be the specific effects from lysoPS (if proved) may in fact be paracrine?

True, paracrine effects due to mediator release from other cell types may contribute to P2Y10-dependent facilitation of chemokine-induced CD4 T cell migration. We used the term “autocrine” mainly for two reasons: First, P2Y10-dependent facilitation of CD4 T cell migration does not require the presence of other cell types (which admittedly also leaves room for paracrine CD4/CD4 communication). Second, exogenous addition of ATP or LysoPS did not affect or even inhibited in all tested conditions of CD4 T cell migration (see also our response to point 8 below), suggesting that a pro-migratory effect of ATP or LysoPS is only observed if the mediators are presented in a certain spatial or temporal context. These findings seemed to us best explained by an autocrine release mechanism, but since we cannot rule out paracrine effects, we replaced the term “autocrine” by “auto-/paracrine” throughout the manuscript. A statement addressing these issues can be found on page 16, lines 14-23. We have also included the possibility of paracrine stimulation in our schematic overview in Figure 7C of the revised manuscript.

8. The authors can further try supplying non-hydrolysable forms of ATP to discern whether endogenous ATP is needed for activating P2Y10, and the ATP aspect of autocrine signaling may also be worked out from these studies?

We are not absolutely certain we understand the question correctly, but assume the reviewer asks for data regarding the effect of exogenous ATP (normal or non-hydrolysable) on CD4 migration. We tested the effect of exogenous ATP (1 μM) on CD4 T cell migration, but did not find effects on basal or chemokine induced migration in control or KO cells (Figure 32A). Also higher ATP concentrations (from 10 μM to 1 mM) did not facilitate migration of CD4 T cells, it had rather opposite effects (as described in literature, for example¹⁴) (Figure 32B). We nevertheless followed the reviewer’s suggestion and tested the effect of non-hydrolysable ATPγS, but again did not observe significant effects on CD4 T cells migration (Figure 32C).

Figure 32: A, Transwell migration of naïve CD4 T cells from control mice and CD4-P2Y10-KOs under basal conditions (left) and in response to CCL19 (100 ng/ml) in the absence and presence of ATP 1 μ M (n=4). **B,C,** Transwell migration of CD4 T cells from control mice in response to increasing concentrations of normal ATP (B, n=2) or non-hydrolysable ATP γ S (C, n=5-6). *, $P < 0.05$; n, number of individual mice per group.

We discuss the finding that exogenous ATP does not facilitate basal or chemokine-induced migration on page 16, lines 16-24.

9. The authors can also try to co-culture LPS stimulated macrophages with naïve CD4 T cells, and see if macrophages stimulation causes stimulation of CD4 T cells for instance?

We followed the reviewer's suggestion and investigated whether LPS-stimulated macrophages modulate CD4 T cell migration. To do so, peritoneal murine macrophages were stimulated over night with 1 μ g/ml LPS, washed twice, and then added together with naïve CD4 T cells to the inserts of 96 well transwell plates (1×10^5 macrophages + 1×10^5 CD4 T cells per insert). Cells were then allowed to migrate in the absence or presence of 100 ng/ml SDF-1 α or 100 ng/ml CCL19 in the bottom well. Except for some inhibitory effect on basal migration, there was no clear effect on CD4 migration (Figure 33). This might be due to the fact that LPS stimulated macrophages can be expected to release a plethora of different mediators, which might affect CD4 T cell migration both positively and negatively or alter adhesive properties of these cells.

Figure 33: Transwell migration of naïve CD4 T cells in the presence or absence of LPS-stimulated macrophages under basal conditions or after addition of 100 ng/ml of the indicated chemokines to the lower well (4 independent experiments with cells from 2 mice).

LysoPS and neuroinflammation:

10. LysoPSs are known to induce neuroinflammation, especially very-long chain components, in the context of the neurological disorder PHARC. Yet, while the authors discuss about lysoPS and neuroinflammation, it is disappointing that the authors don't discuss this pertinent aspect at all.

Following the reviewer's suggestion we included the following statement on PHARC and LysoPS in the discussion on page 15, lines 14-17:

"Mutations in the LysoPS lipase ABHD12 cause the human neurodegenerative disorder PHARC (polyneuropathy, hearing loss, ataxia, retinitis pigmentosa and cataract), and studies in ABHD12-deficient mice suggested that this is due to deregulated LysoPS metabolism and consecutive microglial activation¹⁵"

On this note, it would be good to know whether ABHD12 has any effect on lysoPS in CD4 T cells, and if P2Y10-depletion can rescue neuroinflammation in ABHD12-null mice. ABHD12 inhibitors are described in literature, and the authors can perhaps use these to address this concern.

Following the reviewer's suggestion, we tested the effect of ABHD12 inhibitor DO264 on CD4 T cell migration. The presence of 1 μ M DO264 did not affect the outcome of migration in control cells or KO cells (Figure 34).

Figure 34. Basal and CCL19-induced migration of control and P2Y10-deficient CD4 T cells in the presence and absence of 1 μ M DO264 (added to cells 30 minutes prior to experiment and present throughout the 3 hour migration period) (4 independent experiments with cells from 2 mice per group). *, p<0.05.

We also investigated whether LysoPS levels were altered in chemokine-stimulated CD4 T cells that had been pretreated with DO264 (1 μ M, 1 hr). We did not find significant changes in the

levels of LysoPS 18:0 and 18:1 (Figure 35), suggesting that in CD4 T cells other lipases than ABHD12 can contribute to LysoPS degradation.

Figure 35: Mass spectrometric analysis of LysoPS species 18:0 and 18:1 in control CD4 T cells in the presence or absence of 1 μ M DO264 (n=3-4).

11. Also, lysoPS/ABHD12 has shown to be involved in T-cell mediated LCMV challenges, and it would be good for the authors to discuss this point in the context of their studies as well, and prospect on the role of P2Y10 (and GPR34) in regulating these.

Following the reviewer’s suggestion, we included the following statement regarding the putative role of other LysoPS receptors as regulators of LysoPS-dependent processes on page 15, lines 17-25:

“Also pharmacological inhibition of ABHD12 by DO264 led to elevated LysoPS levels in mouse brain and primary human macrophages and enhanced immunological responses in a mouse model of lymphocytic choriomeningitis virus (LCMV) infection¹⁶. However, these effects are probably not all mediated by P2Y10, since various other GPCRs have been suggested to respond to LysoPS species. The first LysoPS receptor to be deorphanized was GPR34, an X-linked GPCR that is most abundantly expressed in microglia, capable of coupling to G α i-containing heterotrimers, and protective in the central nervous system (CNS) against *Cryptococcus neoformans* infection-induced pathology^{17,18}. Subsequently, GPR174, P2RY10, and P2RY10b, were identified as selective and high-affinity LysoPS receptors using the TGF α shedding screening approach¹⁹.”

Other issues:

- In Figure 1L, Th17 and Th1 were mixed up – this has been corrected in revised Figure.
- The initial labeling of mRNA sequencing data as RPKM was incorrect and has been replaced by the term “library-sized normalized counts” throughout the manuscript

References

1. Han X, Wang R, Zhou Y, Fei L, Sun H, Lai S, Saadatpour A, Zhou Z, Chen H, Ye F, Huang D, Xu Y, Huang W, Jiang M, Jiang X, Mao J, Chen Y, Lu C, Xie J, Fang Q, Wang Y, Yue R, Li T, Huang H, Orkin SH, Yuan GC, Chen M and Guo G. Mapping the Mouse Cell Atlas by Microwell-Seq. *Cell*. 2018;173:1307.
2. Hellberg S, Eklund D, Gawel DR, Kopsen M, Zhang H, Nestor CE, Kockum I, Olsson T, Skogh T, Kastbom A, Sjowall C, Vrethem M, Hakansson I, Benson M, Jenmalm MC, Gustafsson M and Ernerudh J. Dynamic Response Genes in CD4+ T Cells Reveal a Network of Interactive Proteins that Classifies Disease Activity in Multiple Sclerosis. *Cell Rep*. 2016;16:2928-2939.
3. Lee PP, Fitzpatrick DR, Beard C, Jessup HK, Lehar S, Makar KW, Perez-Melgosa M, Sweetser MT, Schlissel MS, Nguyen S, Cherry SR, Tsai JH, Tucker SM, Weaver WM, Kelso A, Jaenisch R and Wilson CB. A critical role for Dnmt1 and DNA methylation in T cell development, function, and survival. *Immunity*. 2001;15:763-74.
4. Heasman SJ, Carlin LM, Cox S, Ng T and Ridley AJ. Coordinated RhoA signaling at the leading edge and uropod is required for T cell transendothelial migration. *J Cell Biol*. 2010;190:553-63.
5. Dubreuil CI, Winton MJ and McKerracher L. Rho activation patterns after spinal cord injury and the role of activated Rho in apoptosis in the central nervous system. *J Cell Biol*. 2003;162:233-43.
6. Berdeaux RL, Diaz B, Kim L and Martin GS. Active Rho is localized to podosomes induced by oncogenic Src and is required for their assembly and function. *J Cell Biol*. 2004;166:317-23.
7. Dybas JM, O'Leary CE, Ding H, Spruce LA, Seeholzer SH and Oliver PM. Integrative proteomics reveals an increase in non-degradative ubiquitylation in activated CD4(+) T cells. *Nat Immunol*. 2019;20:747-755.
8. Uwamizu A, Inoue A, Suzuki K, Okudaira M, Shuto A, Shinjo Y, Ishiguro J, Makide K, Ikubo M, Nakamura S, Jung S, Sayama M, Otani Y, Ohwada T and Aoki J. Lysophosphatidylserine analogues differentially activate three LysoPS receptors. *J Biochem*. 2015;157:151-60.
9. Murphy AC, Lalor SJ, Lynch MA and Mills KH. Infiltration of Th1 and Th17 cells and activation of microglia in the CNS during the course of experimental autoimmune encephalomyelitis. *Brain Behav Immun*. 2010;24:641-51.
10. Tischner D, Grimm M, Kaur H, Staudenraus D, Carvalho J, Looso M, Gunther S, Wanke F, Moos S, Siller N, Breuer J, Schwab N, Zipp F, Waisman A, Kurschus FC, Offermanns S and Wettschureck N. Single-cell profiling reveals GPCR heterogeneity and functional patterning during neuroinflammation. *JCI Insight*. 2017;2.
11. Barnes MJ, Li CM, Xu Y, An J, Huang Y and Cyster JG. The lysophosphatidylserine receptor GPR174 constrains regulatory T cell development and function. *J Exp Med*. 2015;212:1011-20.
12. Shinjo Y, Makide K, Satoh K, Fukami F, Inoue A, Kano K, Otani Y, Ohwada T and Aoki J. Lysophosphatidylserine suppresses IL-2 production in CD4 T cells through LPS3/GPR174. *Biochem Biophys Res Commun*. 2017;494:332-338.
13. Barnes MJ and Cyster JG. Lysophosphatidylserine suppression of T-cell activation via GPR174 requires Galphas proteins. *Immunol Cell Biol*. 2018;96:439-445.
14. Wang CM, Ploia C, Anselmi F, Sarukhan A and Viola A. Adenosine triphosphate acts as a paracrine signaling molecule to reduce the motility of T cells. *EMBO J*. 2014;33:1354-64.
15. Blankman JL, Long JZ, Trauger SA, Siuzdak G and Cravatt BF. ABHD12 controls brain lysophosphatidylserine pathways that are deregulated in a murine model of the neurodegenerative disease PHARC. *Proc Natl Acad Sci U S A*. 2013;110:1500-5.
16. Ogasawara D, Ichu TA, Vartabedian VF, Benthuisen J, Jing H, Reed A, Ulanovskaya OA, Hulce JJ, Roberts A, Brown S, Rosen H, Teijaro JR and Cravatt BF. Selective blockade of the lyso-PS lipase ABHD12 stimulates immune responses in vivo. *Nat Chem Biol*. 2018;14:1099-1108.
17. Liebscher I, Muller U, Teupser D, Engemaier E, Engel KM, Ritscher L, Thor D, Sangkuhl K, Ricken A, Wurm A, Piehler D, Schmutzler S, Fuhrmann H, Albert FW, Reichenbach A, Thiery J, Schoneberg T and Schulz A. Altered immune response in mice deficient for the G protein-coupled receptor GPR34. *J Biol Chem*. 2011;286:2101-10.

18. Kitamura H, Makide K, Shuto A, Ikubo M, Inoue A, Suzuki K, Sato Y, Nakamura S, Otani Y, Ohwada T and Aoki J. GPR34 is a receptor for lysophosphatidylserine with a fatty acid at the sn-2 position. *J Biochem.* 2012;151:511-8.
19. Inoue A, Ishiguro J, Kitamura H, Arima N, Okutani M, Shuto A, Higashiyama S, Ohwada T, Arai H, Makide K and Aoki J. TGAlpha shedding assay: an accurate and versatile method for detecting GPCR activation. *Nat Methods.* 2012;9:1021-9.

REVIEWERS' COMMENTS

Reviewer #1 (Remarks to the Author):

Based on the careful and comprehensive revision, I recommend this manuscript for publication.

Reviewer #2 (Remarks to the Author):

You have answered all my comments thoroughly - I would suggest proof reading the manuscript as there are some grammatical issues with the new insert text.

Reviewer #3 (Remarks to the Author):

The authors have satisfactorily addressed all on my concerns with suitable experiments and explanations. The manuscript is suitable for publication and I congratulate the authors on the nice and exhaustive work.

A minor suggestion to the authors, recently two fairly exhaustive reviews on lyso-PS were published:

1. DOI: 10.1007/s12013-021-00988-9 (Omi, et al. Cell Biochem Biophys, 2021)
2. DOI: 10.1007/s00232-020-00133-2 (Shanbhag et al. J Memb Biol, 2020)

It would be better to reference these reviews that cover this literature instead of multiple individual references regarding this topic in the introduction and discussion sections of the manuscript.